# Estimation of groundwater age distributions from hydrochemistry: Comparison of two metamodelling algorithms in the Heretaunga Plains aquifer system, New Zealand

Conny Tschritter[1], Christopher J. Daughney[2], Sapthala Karalliyadda[3], Brioch Hemmings[1], Uwe Morgenstern[3], Catherine Moore[3]

[1]GNS Science, Taupo, New Zealand
[2]National Institute of Water and Atmospheric Research, Wellington, New Zealand
[3]GNS Science, Lower Hutt, New Zealand

*Correspondence to:* Conny Tschritter (c.tschritter@gns.cri.nz)

## Abstract

Groundwater age or residence time is important for identifying flow and contaminant pathways through groundwater systems. Typically, groundwater age and age distributions are inferred via lumped parameter models based on measured age tracer concentrations. However, due to cost and time constraints, age tracers are usually only sampled at a small percentage of the wells in a catchment. This paper describes and compares two methods to increase the number of groundwater age data points and assist with validating age distributions inferred from lumped parameter models. Two machine learning techniques with different strengths were applied to develop two independent metamodels that each aim to establish relationships between the hydrochemical parameters and the modelled groundwater age distributions in one test catchment. Ensemble medians from the best model realisations per age distribution percentile were used for comparison with the results from traditional lumped parameter models based on age tracers. Results show that both metamodelling techniques predict age distributions from hydrochemistry with good correspondence to traditional LPM-derived age distributions. Therefore, these techniques can be used to assist with the interpretation of lumped parameter models where age tracers have been sampled, and they can also be applied to predict groundwater age distributions for wells in a similar hydrogeological regime that have hydrochemistry data available, but no age tracer data.

## 1.     Introduction

Groundwater age describes the residence time of a parcel of water within the aquifer system, i.e., the time elapsed since recharge. Water from different flow pathways converges at sampling points such as wells and springs. Thus, each sample is a mixture of different groundwater with varying sources and ages (Maloszewski and Zuber, 1996). Understanding the ages of water in the groundwater system is key to determining flow paths, recharge rates and recharge sources, as well as understanding the sustainability of groundwater abstraction, the movement of contaminants in water and the impacts of land use on water quality (Ginn et al., 2009; Daughney et al., 2010; Massoudieh et al., 2012; Morgenstern and Daughney, 2012).

Groundwater age cannot be measured directly but rather must be evaluated using models. There are two main modelling methods used to infer groundwater age distribution and the mean age, or mean residence time. Commonly, groundwater age is estimated through age tracer concentrations, in combination with lumped parameter models (LPMs), to reflect mixing of water from different flow paths. LPMs have also been used to estimate watershed-scale travel time distributions (TTDs) on the basis of hydrogeological information (Abrams and Haitjema, 2018). However, such approaches are unable to resolve the fine-scale spatial variations (heterogeneity) in groundwater age distributions at individual well scales.

LPMs are quick to use and allow the representation of different flow and mixing models. They can be matched to the measured concentrations of environmental tracers, like tritium, sulphur hexafluoride ($SF_6$) and chlorofluorocarbons (CFCs), in the groundwater sample (Maloszewski and Zuber, 1996; Morgenstern and Daughney, 2012). Tracer-informed LPM age interpretations are made on a site-by-site basis. One particular disadvantage is that these types of LPMs can only be fitted to

locations at which tracer concentrations have been measured, and can only represent the aggregation of a heterogeneous system along the groundwater flow path, rather than detailed flow path variability.

As an alternative to LPMs, physically based numerical flow and transport models can also be used to assess groundwater age and transit time distributions. Some such investigations have focused on the mathematical descriptions of groundwater age and its dynamics (Ginn et al., 2009; Cornaton, 2012; Engdahl, 2017), whereas other investigations have evaluated the role of groundwater age and environmental tracer data in model calibration, alongside other data sources such as hydraulic heads and stream flow measurements (Portniaguine and Solomon; 1998; Zhu, 2000; Massoudieh et al., 2012). An advantage of the numerical modelling studies is that they can evaluate age distributions spatially and temporally across the entire model domain, and account for age distributions with more complex shapes than can be represented by simple LPMs. An effect of deriving age distributions from a spatiotemporally modelled groundwater flow field is that there is likely to be some correlation and possible carry-over of biases in between the age interpretations for sites that are near to each other, which is less likely for LPM-derived ages. A key disadvantage is that the development of numerical models typically requires much more time and effort compared to the simpler LPMs, even after accounting for the time and costs of measuring the environmental tracer concentrations at the sites of interest. Additionally, as outlined in Knowling et al. (2020), numerical models require appropriate structure and parameterisation to ensure that the information from age tracers can be robustly assimilated by the model.

In recognition of the limitations of the above-listed methods, various less time- and cost-intensive methods have previously been trialled to increase the amount of available groundwater age data in areas where no age tracers have been sampled and analysed, and where a numerical flow and transport model is not available. Typically, these alternative methods for estimating groundwater age rely on groundwater chemistry data, hydrogeological information (e.g. bore construction parameters, recharge, geology, etc.), or a combination thereof (Edmunds and Smedley, 2000; Daughney et al., 2010; Beyer et al., 2016; Marçais et al., 2018), linking groundwater chemistry and hydrogeological parameters to groundwater age and transit time of water through the aquifer. Most such previous studies have relied on statistical data analysis methods, i.e. discriminant analysis, principal component analysis, regression analysis etc., that were used independently or in combinations with each other to identify and model relationships between groundwater chemistry and age data (Daughney et al., 2010; Beyer et al., 2016). These methods have been shown to be reasonably successful in deriving mean groundwater age, either as an age category or absolute age, but did not provide estimates of the full groundwater age distributions, which are more meaningful for contaminant transport and drinking water security than mean age (Beyer et al., 2016; Weissmann et al., 2002; Suckow, 2014). This study builds on previous investigations of the use of groundwater chemistry as a proxy to infer groundwater age, with the aim of using metamodels to assess the full age distribution instead of just the mean age. Metamodels (also known as 'surrogate' or 'data-driven' models) are statistical or machine learning-based 'models of models' that can be used to extrapolate relationships to enable predictions to be made at unsampled locations or times; metamodels are thus models that are trained on other models that themselves had been previously calibrated on observed data (Fienen et al., 2015, 2016, 2018; Asher et al., 2015; Starn and Belitz, 2018; Starn et al. 2021). Therefore, metamodels provide a cost-efficient alternative to both physically based distributed numerical models or LPMs, whenever sufficient training data exists (Razavi et al., 2012). Alternatively, in more data-sparse contexts, they may be used in combination with numerical modelling efforts (Koch et al., 2019; Reichstein et al., 2019). Metamodels can make relatively rapid predictions of system behaviour or characteristics based on the relationships that are established with observed data. Although metamodel predictions will typically have a higher uncertainty than numerical model predictions (due to the fact that they are trained on models which have their own uncertainties), they can be made more rapidly while efficiently dealing with high parameter dimensionality (Fienen et al., 2016). Metamodels have been developed for various hydrogeological applications (Fienen et al., 2018; Nolan et al., 2018; Starn and Belitz, 2018; Asher et al., 2015), including the prediction of groundwater age distributions from hydrogeographic and bore-specific observations and features, or numerical flow model outputs (Fienen et al., 2016; Starn and Belitz, 2018).

However, none of these metamodelling studies have investigated the use of hydrochemistry for the predictions of groundwater age.

Specifically, this study evaluates and compares the performance of two ensemble machine learning techniques (symbolic regression and gradient boosted regression) with the goal of estimating groundwater age distributions from groundwater chemical composition. Symbolic regression (SR) is a machine learning technique that attempts to identify explicit mathematical expressions in an input dataset. It is initiated by developing a population of naïve random mathematical expressions that conform to *a priori* selected grammar rules. The initial mathematical expressions are then combined and evolved through an approach such as genetic programming, to develop a set of formulas that describe the relationship(s) of interest with sufficient accuracy (Gomes et al., 2019). Gradient boosted regression (GBR) is a machine learning method that aims to minimize the prediction error through a regression tree model - a sequence of regression trees. Each sequential addition of the new regression tree will minimize the prediction error made by the previous tree and thus decrease the overall prediction error. Whilst there are numerous possible machine learning methods that can be used for this purpose (e.g. Random Forest, Bagged decision trees, Neural Networks), we selected the SR and GBR techniques based on the amount of available data, ease of use and adaptability, and/or proven potential in similar research. For example, unlike most other machine learning methods, SR provides the actual equation for the resulting model. This means that the user can directly see the calculation that is being performed, which in turn helps to check on the physical basis of the equation and helps with transferring the model into other software like Microsoft Excel, making it more accessible to a wider user group. GBR, on the other hand, is a highly adaptive, strong predictive model and has previously successfully been used in other studies to predict groundwater age from hydrophysical parameters (Fienen et al., 2018, 2016).

The GBR and SR approaches were implemented to estimate selected percentiles in the LPM groundwater age distribution based on measured groundwater chemistry on a per-sample basis in a test catchment, the Heretaunga Plains, New Zealand. Although this study uses LPM-derived age distributions as the metamodel prediction targets, we note that our approach would also be applicable to the use of groundwater chemistry to predict the age distributions derived from a physically based numerical model, which is an additional research direction being pursued by our group to be reported elsewhere. As noted above, one potential advantage to using LPM-derived age estimates as inputs to metamodelling is that errors may be stochastic rather than systematic in nature compared to the potential for site-to-site correlations in errors or biases in age estimates derived from numerical flow and transport model.

## 2.    Study Area

The Heretaunga Plains is a 300 km$^2$ SW-NE trending fault-bounded depression located on the east coast of New Zealand's North Island (Fig. 1). The Ngaruroro, Tutaekuri and Tukituki are the three main rivers that traverse the Heretaunga Plains, which have long-term, median outflows of 19.9, 8.5 and 21.8 m$^3$/s, respectively (Waldron et al., 2019). The area has a temperate climate with average temperatures of 17°C in summer and 10°C in winter, and average annual rainfall of approximately 800 mm near the coast (Dravid and Brown, 1997). Land cover in the western portion of the catchment is comprised primarily of native forest, scrub, and tussock, whereas the eastern portion is primarily exotic grassland (mostly used for grazing sheep and beef cattle) with lesser areas of orchard, vineyard, and short-rotation cropland, along with urban areas  (Smith et al., 2020).

### 2.1.    Geology and hydrogeology

Starting in the Miocene, tectonic activity associated with the Hikurangi Trough, which is part of the Australian-Pacific plate boundary, resulted in the development of an actively subsiding syncline ('Napier Syncline') in the Heretaunga Plains (Fig. 1). The axis of this syncline is oriented subparallel to the orientation of the lengths of the plains, and the resulting depression has since been infilled by marine and alluvial deposits representing several glacial – interglacial cycles and associated sea level

fluctuations (Begg et al., 2022; Lee et al., 2014). The total depth of this depression is uncertain, but it has been estimated to be between 900 m (Dravid and Brown, 1997) and 1,600 m (Beanland et al., 1998).

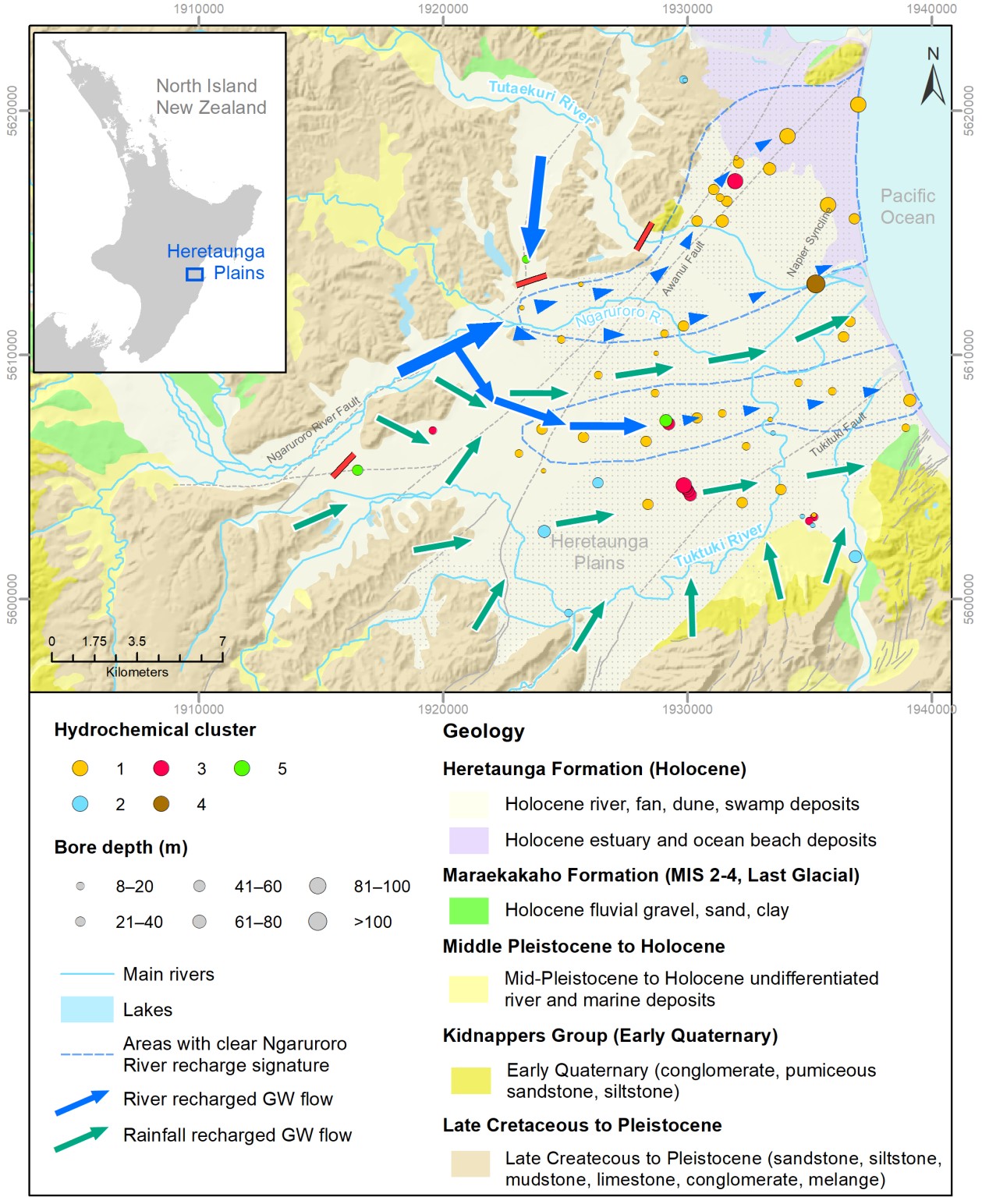

**Figure 1: Bore locations and depth, hydrochemical cluster, geology, and inferred groundwater flow dynamics in the Heretaunga Plains (hydrochemical cluster and groundwater flow dynamics from Morgenstern et al., [2018]; geology from Heron, [2020]; rivers and lakes from LINZ, [2022]). The length of the arrows is proportional to the estimated flow rate. The red lines identify areas where Morgenstern et al. (2018) found indication that there is no surface water flow contributing to the main aquifers. The stippled area shows the extent of fine (sand, silt, clay) estuarine and terrestrial deposits mapped at the ground surface (Lee et al. 2020).**

The main aquifers of the Heretaunga Plains are composed of highly transmissive, gravel-dominated fluvial deposits from the late Pleistocene (Maraekakaho Formation) and Holocene (Heretaunga Formation), deposited by the three major rivers in the plains (Dravid and Brown, 1997). Lee et al. (2014) analysed 4051 lithological well logs provided by regional authorities and

found that most of the primarily 20 – 50 m deep bores terminate in gravels deposited during the last glaciation (71,000–12,000 years ago). Towards the coast, these gravel deposits are overlain by silt- and clay-dominated marine sediments, deposited during the Holocene marine transgression, which thicken towards the coast and act as a confining layer. Smaller gravel aquifers also occur at the coast. Further inland, Holocene terrestrial deposits, i.e., gravel, sand, clay, and silt, interfinger with the marine deposits, resulting in an interconnected confined-unconfined aquifer system (Dravid and Brown, 1997). Thick Holocene gravel fans, associated with the Ngaruroro, Tukituki and Tutaekuri rivers, which have been mapped from bore logs, are likely hydraulically connected to the underlying Last Glacial gravels (Lee et al., 2014; Begg et al., 2022).

Underlying the Pleistocene gravel deposits are Late Cretaceous to Pleistocene marine and terrestrial deposits (mudstone, melange and mudstone, sandstone, siltstone, limestone, and conglomerates). Based on their mapped occurrence outside of the Heretaunga Plains and seismic reflection data from within the plains, these deposits are expected to underlie the study area at depth. However, none of the groundwater bores reach these deposits, and the only bore data available is from a small number of petroleum exploration bores (Dravid and Brown, 1997; Lee et al., 2014).

## 2.2. Groundwater flows

Sources of groundwater recharge into the Heretaunga Plains aquifers have been inferred from river flow gauging surveys (Wilding, 2018), groundwater level monitoring (Smith et al., 2020), numerical modelling (Rakowski, 2018; Rakowski and Knowling, 2018), and assessments of water chemistry, stable isotopes, and age tracers (Morgenstern et al., 2018). These methods collectively indicate that losses from the main rivers occur in limited areas but contribute about two thirds of the total volume of groundwater recharge to the aquifer system (approx. 264 M $m^3$/year; Rakowski & Knowling 2018), with the remainder of recharge sourced from rainfall percolation through the soil zone across a wider area of the Heretaunga Plains (Fig. 1).

The dominant groundwater flow direction is from west to east, following the topographic gradient towards the coast (Fig. 1). Artesian and sub-artesian conditions are observed in bores in the confined aquifer zone near the coast (Dravid and Brown, 1997). Age tracer measurements indicate relatively rapid horizontal groundwater velocities of ca. 3-5 km/year in some parts of the Heretaunga Plains aquifer system, particularly in proximity to losing reaches of the main rivers (Morgenstern et al., 2018). Bores as deep as 75 m below ground surface can have tritium concentrations similar to modern rainfall, indicating that vertical groundwater flow can also be relatively rapid in some areas. In contrast, the older groundwaters and slower flow velocities of ca. 0.1-0.2 km/year are inferred nearer the coast, which could result from widening of the aquifer cross section and/or decreasing hydraulic conductivity, e.g., reflecting the presence of finer-grained sediments of marine origin (Morgenstern et al., 2018).

Approximately 40% of the discharge from the aquifer system is estimated to occur via seepage into streams and springs, with the remaining discharge evenly split between abstraction and flows across the coastal boundary (Rakowski and Knowling, 2018). Total abstraction has approximately doubled in the last 30 years, with an average annual increase of approximately 3.5%, due primarily to increases in irrigation and industrial use of groundwater (Rakowski and Knowling, 2018). This increase in total abstraction is inferred to be the cause of long-term declines of summer groundwater levels (average rate ca. 5 cm/year between 1989 and 2018), which are observed in some unconfined parts of the aquifer system (Smith et al., 2020).

## 2.3. Hydrochemistry

Groundwaters in the Heretaunga Plains have a range of hydrochemistry (Fig. 1), arising from the spatially variable processes of human impact and natural geochemical evolution, as observed elsewhere in New Zealand (Daughney et al., 2012; Morgenstern and Daughney, 2012). Generally, natural geochemical evolution is expected to affect the redox state, with younger groundwaters more likely to be oxic than anoxic, thereby affecting the concentrations of redox-sensitive substances such as dissolved oxygen (DO), $NO_3$-N, $NH_3$-N, Fe, Mn and $SO_4$ (Tesoriero and Puckett, 2011; Daughney et al., 2010) (see

list of chemical abbreviations and units in the Supplementary Material Table S1). Natural water-rock-interaction also typically causes the concentrations of the major ions to increase with time and distance along a groundwater flow path (Morgenstern and Daughney, 2012). Human influence on groundwater chemistry in New Zealand is primarily indicated by elevated concentrations of $NO_3$-N, sometimes co-occurring with elevated concentrations of Na, K, Mg and/or Cl (Daughney et al., 2012; Morgenstern and Daughney, 2012). The dominant recharge source also influences hydrochemistry, with groundwaters sourced primarily from rainfall seepage through the soil zone tending to have higher total dissolved solids (TDS) and higher concentrations of the parameters associated with human activity compared to groundwaters sourced from river seepage (Morgenstern and Daughney, 2012). These general drivers of hydrochemistry can lead to reasonably strong correlations among the levels of several parameters, as is observed for the groundwaters in the Heretaunga Plains (Fig. 2). The following paragraphs summarise the key correlations and patterns among the hydrochemical variables based on facies identified by hierarchical clustering as previously reported by Morgenstern et al. (2018).

Oxic groundwaters inferred to be recharged from rivers are found across much of the study area (denoted as Cluster 1 in Fig. 1). These groundwaters typically have Ca and $HCO_3$ as the dominant cation and anion, with concentrations of ca. 20-30 mg/L and 50-100 mg/L, respectively (Morgenstern et al., 2018). Due to their redox status, such groundwaters have concentrations of DO, $NO_3$-N and $SO_4$ above their respective analytical detection limits, but concentrations of Fe, Mn and $NH_3$-N are usually below detection. These groundwaters display relatively little indication of land-use impacts: concentrations of $NO_3$-N are typically below 1 mg/L, and microbial pathogens and pesticides are generally not detected (Smith et al., 2020). In some locations, particularly near the margins of the plains, these river-recharged groundwaters can display concentrations of Ca and $HCO_3$ that are 2-3 times higher than elsewhere, likely due to the influence of carbonate-rich geologies in the surrounding hills (denoted as Cluster 2 in Fig. 1).

Oxic groundwaters inferred to be recharged from rainfall occur in a small number of areas of the plains (denoted as Cluster 3 in Fig. 1). These groundwaters also typically have $HCO_3$ as the dominant anion but can have either Ca or Na as the dominant cation (Morgenstern et al., 2018). Otherwise, these groundwaters are generally hydrochemically similar to the oxic river-recharged groundwaters described above, except for having slightly higher concentrations of $NO_3$-N, typically in the range 2-2.5 mg/L, as a result of modest land use impacts, along with slightly higher concentrations of Ca, Mg, Na, K and/or $SiO_2$ due to their accumulation during passage of recharge water through the soil zone (see Daughney and Morgenstern, 2012).

Anoxic groundwaters occur in a small number of wells (denoted as Cluster 4 in Fig. 1). Depending on their redox state, these groundwaters typically have detectable concentrations of Fe, Mn and/or $NH_3$-N but low or non-detectable concentrations of DO, $NO_3$-N and/or $SO_4$. Concentrations of $PO_4$-P are also observed to be higher in anoxic than oxic groundwaters, likely due to solubilisation associated with reductive dissolution of iron oxide minerals in the aquifer (Langmuir, 1997). A small number of wells have $NH_3$-N concentrations roughly twice as high as elsewhere, which may indicate contamination by wastewater (denoted as Cluster 5 in Fig. 1).

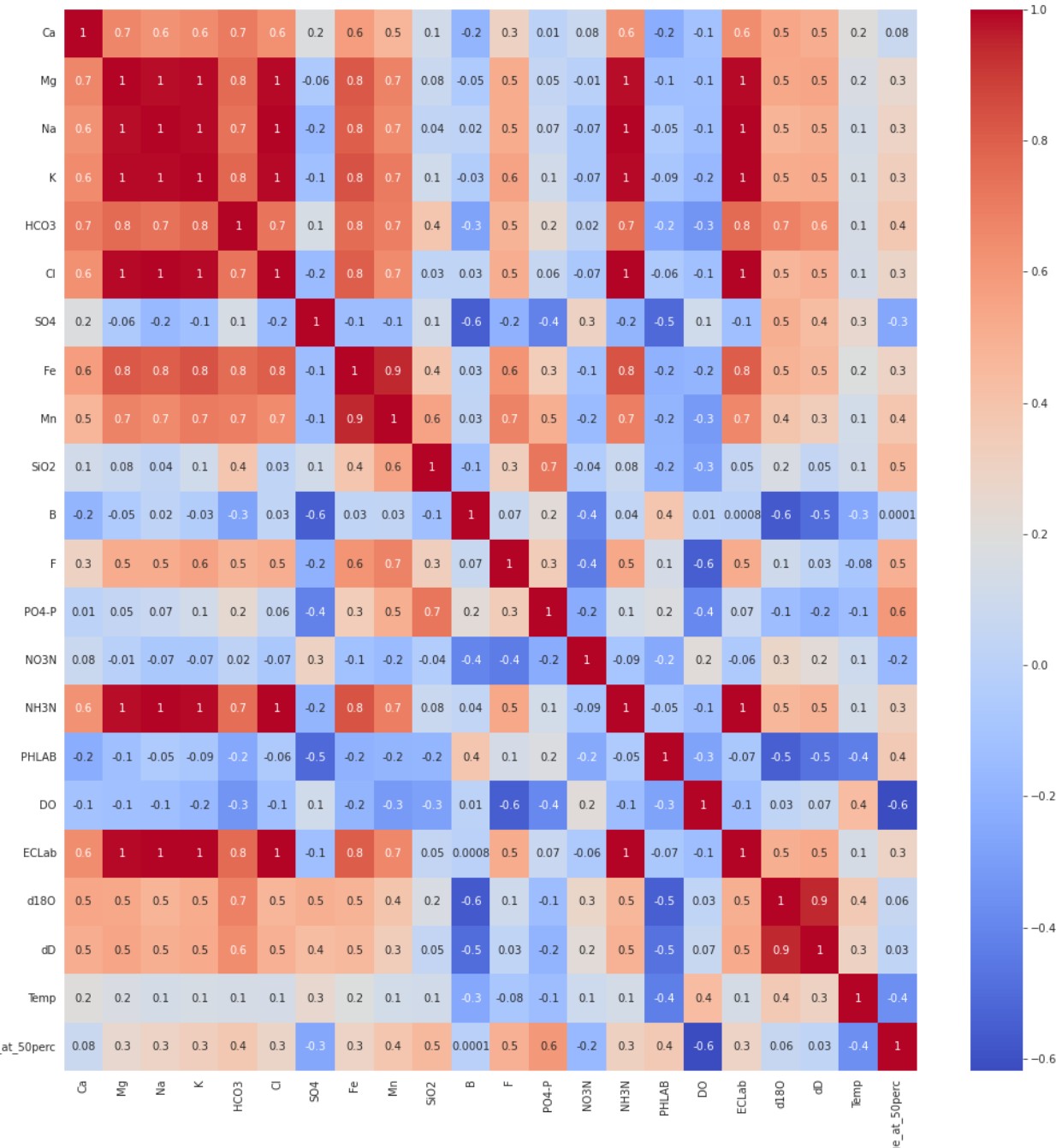

**Figure 2: Pearson R correlation matrix among hydrochemical parameters and groundwater age, estimated via the LPM at the 50th percentile of the age distribution, across all Heretaunga Plains groundwater samples used in this study (n=76).**

Groundwater chemistry shows complex relationships to groundwater age in the Heretaunga Plains (Morgenstern et al., 2018). Generally, the median value of mean residence time for shallow groundwaters within the Holocene unconfined gravels is estimated to be between zero and ten years, with a progressive increase to the range 40-80 years, for deeper groundwaters, near the coast (Fig. 3).

Younger groundwaters are more likely to be oxic, whereas deeper groundwaters are more likely to be anoxic, which affects the location and depth profiles of DO, $NO_3$-N, $NH_3$-N, Fe, Mn, $SO_4$ and $PO_4$-P. However, due to the complex flow paths in the Heretaunga Plains aquifer system, the relationships between age, location, depth, and groundwater chemistry are also complex, for example there are locations where young groundwaters are found at depth and older groundwaters are found near the surface. Accordingly, Morgenstern et al. (2018) did not report any predictive relationships between groundwater chemistry and groundwater age. Moreover, the relationships among the redox-sensitive hydrochemical parameters such as DO, $NO_3$-N,

NH3-N, Fe, Mn and SO4 are themselves known to be non-linear because they are mediated by step-wise microbial respiration reactions (Langmuir, 1997; McMahon and Chapelle, 2008).

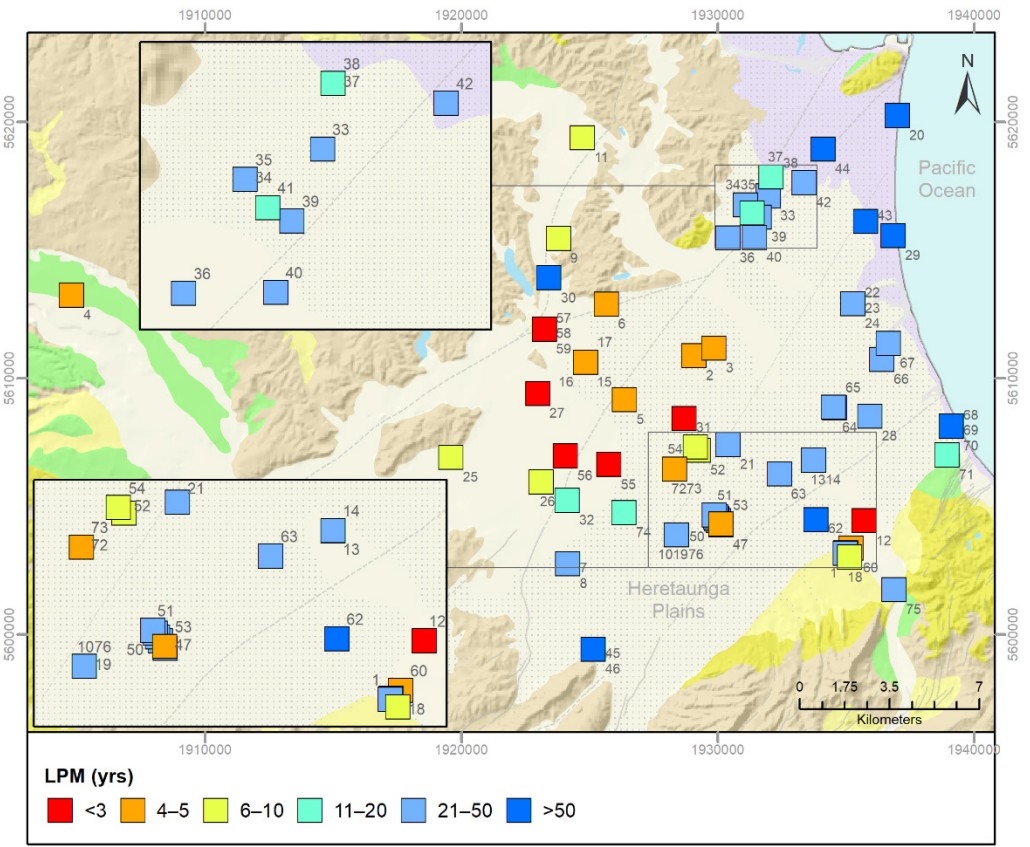

**Figure 3: Locations of sites used for the model development, showing the LPM-derived ages for the 50th percentile. The geological formations shown on the background map (Heron, 2020) are explained in Fig. 1. Labels correspond to mapIDs for each site.**

## 3.  Methods

### 3.1.  Data

This study used hydrochemical data from Morgenstern et al. (2018) (Supplementary Material Table S3) as predictor variables for the modelling approaches. The dataset is comprised of 76 groundwater samples collected from 69 sites in the Heretaunga Plains (Fig. 1). Bore depths ranged from 8 to 147 metres below ground surface (m bgs) (25th, 50th and 75th percentiles were 30, 46 and 71 m bgs, respectively). These hydrochemical samples were mainly collected during sampling campaigns in 2014, 2016 and 2019. Most samples (75%) were collected in the period April to June, with approximately even proportions of the remaining samples collected in the periods January to March or November to December. All sites were sampled according to standard protocols involving purging of bores and stabilisation of pH, DO, electrical conductivity (EC) and temperature (T) as measured in the field using portable meters prior to sample collection (Daughney et al., 2007). Censored and uncensored results below the highest censoring threshold for each parameter were replaced with the corresponding analytical detection limit (Helsel et al., 2020) in order to allow application of the machine learning methods in this study.

As response variables for the modelling, this study used groundwater age distributions based on age tracer data from Morgenstern & van der Raaij (2019). The age tracers, tritium ($^3$H), CFCs and SF6 were selected for their appropriateness for the relatively young groundwaters found in many New Zealand aquifer systems (Stewart and Morgenstern, 2001). One set of age tracer samples was collected from each site at the same times as the above-mentioned samples that were analysed for hydrochemistry. Additionally, at 39 of the 76 sites, between two and 12 additional sets of age tracer samples had been collected for other investigations extending back as early as 1995. We point out that, in New Zealand and the Southern Hemisphere in general, bomb tritium has now fully dispersed, reducing ambiguity when fitting LPMs and increasing the reliability of tritium-

based age interpretations. This is not yet the case for the Northern Hemisphere, where bomb tritium is still present in significant

amounts within the groundwater systems and causes ambiguity in age interpretations from tritium (Stewart et al., 2021).

All age tracer analyses were performed at the GNS Science Water Dating Lab. Tritium was analysed in a 1 L unfiltered, unpreserved sample using 95-fold electrolytic enrichment followed by ultra-low-level liquid scintillation spectrometry (Morgenstern and Taylor, 2009). The detection limit was 0.02 tritium units (TU), and the reproducibility of a standard

enrichment was 1% via deuterium calibration. Samples for analysis of CFCs and $SF_6$ were collected in strict isolation from the atmosphere, as described by Daughney et al. (2007), using 125 mL and 1 L bottles, respectively. Concentrations of CFCs (CFC-11 and CFC-12) and $SF_6$ were analysed at GNS Science by gas chromatography (GC) using an electron capture detector as described by Busenberg and Plummer (1992) and van der Raaij (2003). Detection limits were $3\times10^{-15}$ mol $kg^{-1}$ for CFCs and $2\times10^{-17}$ mol $kg^{-1}$ for $SF_6$. Dissolved argon and nitrogen concentrations were measured simultaneously with CFCs by GC

using a thermal conductivity detector (analytical accuracy is 1% and 3%, respectively). The argon and nitrogen concentrations were used to estimate the temperature at the time of recharge and the excess air concentration as described by Heaton and Vogel (1981), which allowed calculation of the atmospheric partial pressure (ppt) of CFCs and $SF_6$ at the time of recharge.

This study made use of all available age tracer data to constrain the LPM for the relevant site. Use of data from several different tracers allows their applicable age ranges and behaviours in the aquifer system to be accounted for, enabling derivation of the

most robust LPM interpretation consistent with them all (Stewart and Morgenstern, 2001). Evaluation of the groundwater age distribution involved fitting of a LPM to the age tracer data using the TracerLPM workbook (Jurgens et al., 2012), following the approaches of Daughney et al. (2010) and Morgenstern et al. (2015). This involved use of the convolution integral to compare the measured tracer concentration at the sampling point ($C_{out}$) with its concentration in rainfall at the time of recharge ($C_{in}$), calculated following Eq. (1):

$$C_{out}(t) = \int_0^\infty C_{in}(t-\tau)\, e^{-\lambda t} g(\tau) d\tau \tag{1}$$

where $t$ is the time of observation, $\tau$ is the transit time (groundwater age), $e^{-\lambda\tau}$ is the decay term with $\lambda = \ln(2)/T_{1/2}$ (i.e. radioactive decay term for tritium with half-life $T_{1/2} = 12.32$ years) and $g(\tau)$ is the system response function (Zuber et al., 2005). The time-series $C_{in}$ for tritium input via rainfall was based on concentrations measured monthly at Kaitoke, near Wellington, New Zealand, since the 1960s (Morgenstern and Taylor, 2009), whereas the time-series for inputs of CFCs and

$SF_6$ were based on measured and reconstructed data from Cape Grim, Australia, and other southern hemisphere sites (Cunnold et al., 1997; Maiss and Brenninkmeijer, 1998; Prinn et al., 2000; Thompson et al., 2004). The system response function defines the shape of the distribution of ages within the water sample, for example, as arising from convergence and mixing of groundwater flow paths at the well during sampling. System response functions comprised by a singular or binary exponential piston flow model (EPM) have been shown to provide good matches to time-series age tracer data for a wide range of New

Zealand groundwater systems (Daughney et al., 2010; Morgenstern and Daughney, 2012; Morgenstern et al., 2015), including in the Heretaunga Plains, a groundwater system with an unconfined zone upgradient and a confined zone downgradient (Morgenstern et al. 2018; Morgenstern & van der Raaij, 2019). A singular EPM involves estimation of two parameters , T and f:

$$g = 0 \text{ for } \tau < T(1-f) \tag{2}$$

$$g = \frac{1}{Tf} e^{\left(-\frac{\tau}{Tf} + \frac{1}{f} - 1\right)} \text{ for } \tau \geq T(1-f) \tag{3}$$

where T is the mean residence time (MRT) and f is the ratio of the volume of exponential flow to the total flow volume at the groundwater discharge point, with T(1-f) being the time it takes for groundwater to flow through the piston flow section of the aquifer. A binary EPM combines two singular EPMs and hence has five unknowns: $T_1$ and $f_1$ for the first EPM, $T_2$ and $f_2$ for the second EPM, and r, the ratio of the two single EPMs in an overall system response function used to model the final water

age distribution. For six of the sites considered in this study, the age tracer data deviated significantly from any single LPM

derived model (regardless of whether a EPM or BMM lumped parameter model was adopted). This deviation is assumed to represent sampling of dynamic flow behaviour, or a temporally evolving or changing system. For the purposes of the metamodelling, these data were treated as separate samples, with unique chemical and LPM derived age signatures, in the input dataset.

### 3.2. Symbolic Regression and Gradient Boosted Regression Models

SR and GBR models were developed using data from all sites in a single group; in other words, sites were not pre-segregated into different groups based e.g. on hydrochemical cluster, well depth, or any other characteristic. While we acknowledge that pre-segregation of input data followed by development of separate machine learning models is used in some studies, one goal of our investigation was to determine whether the SR and GBR algorithms could themselves account for any inherent differences in the age-chemistry relationships between sites, without pre-segregation. The effect of this approach is discussed in Sect. 4.1.

SR models were developed using HeuristicsLab version 3.3.16.17186 (Wagner et al., 2014). SR settings allowed a maximum tree depth and length of 15 and 150, respectively, based on a multi-symbolic expression crossover with internal crossover point probability of 90%. SR grammar rules permitted arithmetic, exponential and logarithmic functions; permission of conditionals (e.g. if/then statements) was also assessed in terms of ability to improve model fits.

GBR models were developed using the GBR package that is available with the open-source scikit-learn library in Python (Pedregosa et al., 2011). The hyperparameters were tuned to find the optimal parameters (tree depth = 4, sample split = 2 and learning rate = 0.05) that result in the best performance of the models. A stopping criterion was applied to determine the number of estimators (regression trees) required (if the model score was not improved by at least 0.01 in the last 50 iterations then the model was considered to have converged) and, in most of the cases, the models achieved their optimal solution at around 50 –75 estimators (boosting iterations).

The first stage in developing the SR and GBR models was to generate an ensemble of independent models for each of nine selected percentiles ($5^{th}$, $10^{th}$, $20^{th}$, $33^{rd}$, $50^{th}$, $66^{th}$, $80^{th}$, $90^{th}$, $95^{th}$) in the LPM-derived water age distributions. Hereafter these are referred to as 'unchained models' to differentiate them from the 'chained models' described below. The input dataset for the unchained models consisted of the sample-specific values for 21 hydrochemical parameters: Ca, Mg, Na, K, $HCO_3$, Cl, $SO_4$, Fe, Mn, $SiO_2$, $NO_3$-N, $NH_3$-N, $PO_4$-P, pH, EC, $\delta^2H$ and $\delta^{18}O$ (all measured in the lab), along with T and DO (measured in the field). The purpose of developing these unchained models for individual percentiles was to enable testing the validity of the shapes of the age distributions produced by the LPMs. For example, a site with an unrealistic LPM-derived age distribution might be identified by unchained SR or GBR models that perform well for some percentiles but not others.

For both the SR and GBR methods, for each age percentile, ten data split realizations were generated by dividing the input data into testing and training subsets. We tested a range of test/train split ratios for each method, based on typical approaches used by practitioners use of these modelling methods. One hundred repeat models were constructed for each split realization. For the SR method, each split realization was constructed through independent and random division of ten input data duplicates, with a test/train split ratio of 33/66 found to deliver good stability for metamodel development. For the GBR method, the input data was divided into ten folds in a 10-fold cross-validation procedure, and a test/train split ratio of 10/90 was determined to be optimal. In the cross-validation procedure, each fold was sequentially "held-out" in the testing data with the remaining nine folds comprising the training data set; this was repeated 100 times with some shuffling of the data between folds, for each repeat. Then, for both SR and GBR, from the total of 100 models produced at each split realization or fold, we selected the four best performing models. The best performing models were defined as those with the highest Pearson $R^2$, as long as $R^2_{Training} \geq 0.7$ and $ABS(R^2_{Training} - R^2_{Testing}) \leq 0.2$, ensuring reasonable and similarly high correspondence to the LPM-derived training and testing datasets, analogously to the Akaike Information Criterion (Gomes et al., 2019). We note that these criteria do not discuss model performance beyond model-to-measurement fits, and that selection of such criteria may result in

biased model rankings, as discussed by Schöniger et al. (2014); however, these criteria are commonly adopted for data-driven methodologies.

Overall, for both SR and GBR, this approach produced a final group of 40 independent models for each of the nine above-listed age percentiles (Fig. 4). These resulting model ensembles were summarized using the average, median, median absolute deviation (MAD), and standard deviation (SD) of the predictions for each of the nine percentiles in the age distributions. The SR and GBR methods also automatically determined the influence of each of the above-listed input variables with respect to model predictions for each age percentile, providing a quantification of the relative importance ('feature importance') of each

input variable for the prediction of groundwater age distributions. This feature importance, derived from the unchained models, was subsequently used to provide insights into the physical and chemical processes that characterise the specific hydrogeological system.

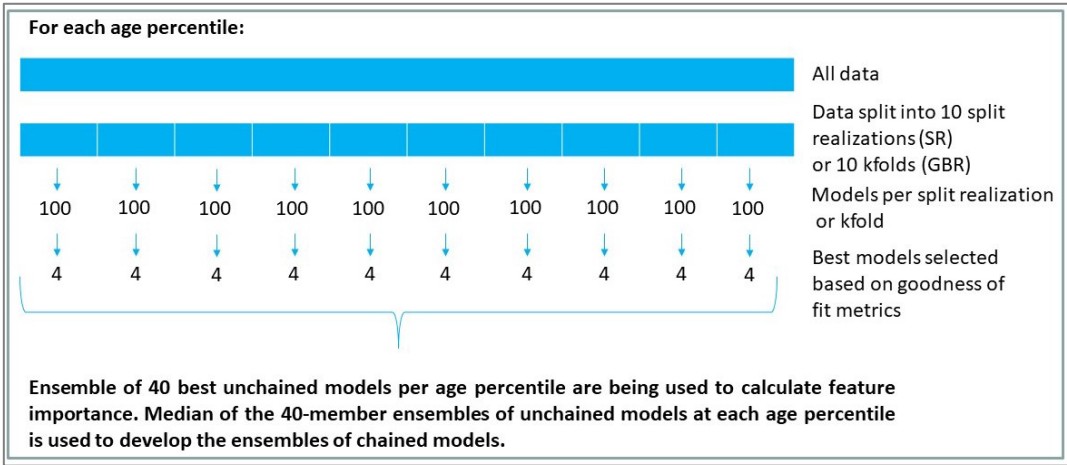

**Figure 4: Schematic of workflow used for SR and GBR modelling.**

The second stage in developing the SR and GBR models was to implement a chaining approach that connected the models for the unchained percentiles in the age distributions. This was done to ensure that the separately simulated percentiles had an appropriate relationship to each other, e.g., that the value for the 10th percentile in the age distribution for any sample had to be greater than or equal to the 5th percentile in the age distribution at the same sample. The implementation of the chaining approaches for the SR and GBR models varied slightly. For the SR method, independent models were first developed for each

of the nine percentiles in the age distribution as described above, then the model for each individual percentile was re-modelled based on the ensemble median value from all age percentiles; for example, the chained model estimate for the 5th percentile in the age distribution was based on the unchained models for all nine percentiles. For the GBR method, the first step was to use the hydrochemical data to develop an unchained model to simulate the 5th percentile in the age distribution across all samples, as described above. Then this model for the 5th percentile in the age distribution was subsequently used as input, along with

the hydrochemical data, to develop a second model to simulate the 10th percentile in the age distribution across all samples, which in turn was used in conjunction with the hydrochemical data to develop a third model to simulate the 20th percentiles across all sites, and so on. For both the SR and the GBR approaches, the chained model development followed the same split and validation procedure as were used for the development of the unchained models.

To demonstrate an example application, groundwater age distributions were subsequently predicted for sites for which time-

series hydrochemistry data was available, but no tracer data, and therefore, no LPM-derived groundwater age distributions. While, by definition, no assessment of correspondence between metamodel derived age distributions and LPM derived distributions can be made for these sites, the example application to sites with no tracer data serves to demonstrate the use of these metamodelling approaches and does allow comparison of the correspondence between the two approaches.

## 4.    Results and Discussion

This section is split into results and discussions focussing on 1) the comparison of the model predictions made by the two metamodelling approaches and how they correspond with those derived from the LPM model, 2) the relationships between hydrochemistry and groundwater age distributions that are affecting the predictions, and 3) potential applications these techniques could be used for.

### 4.1.    SR and GBR model predictions

Age distribution predictions derived from the SR and GBR trained metamodels are presented in Fig. 5 (for mapped median ages), Figure 6 and Supplementary figures S1-S4 (for age distributions).

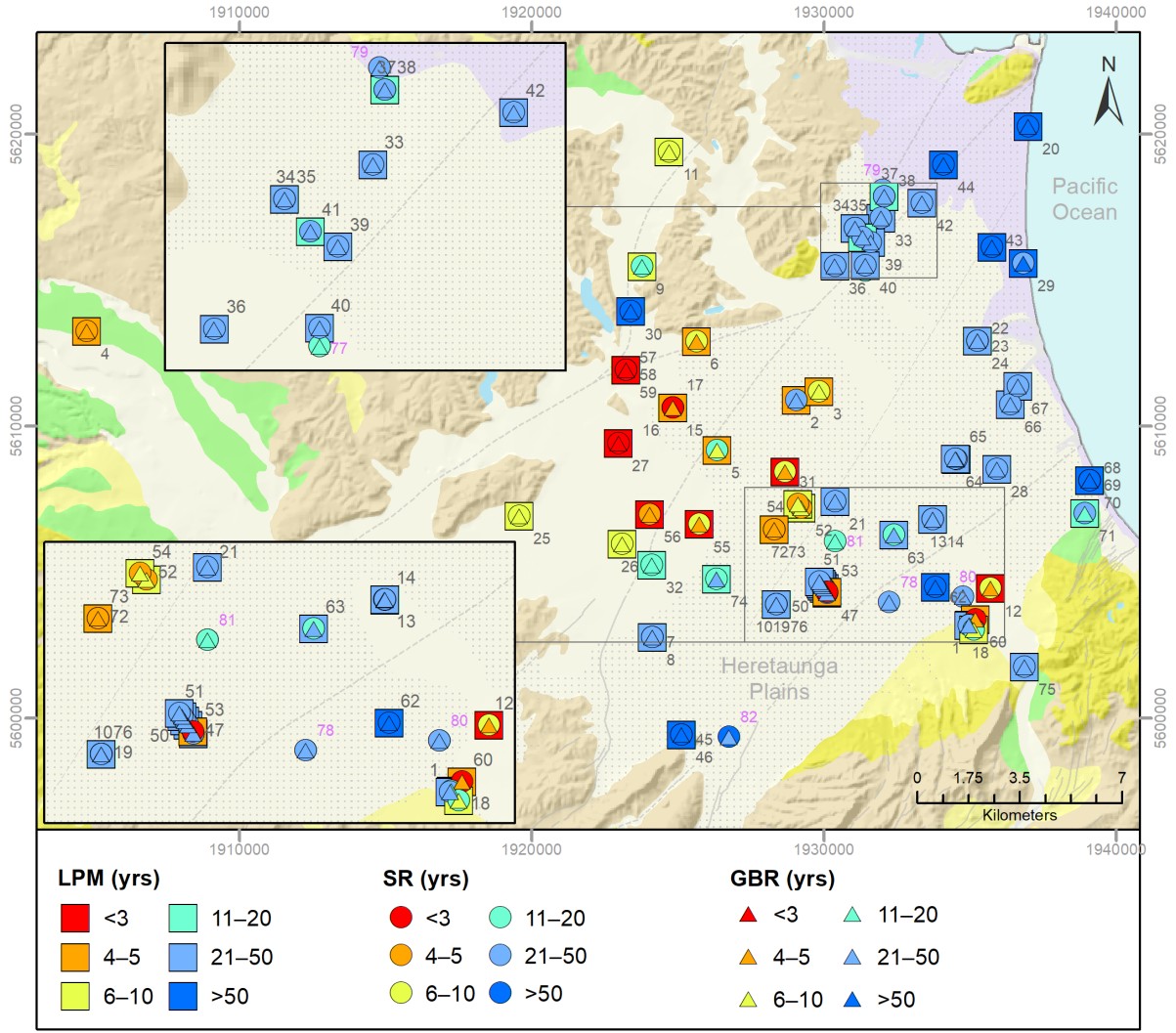

**Figure 5: Comparison between the LPM, and the SR and GBR modelled ages for the 50th percentile. The geological formations shown on the background map (Heron, 2020) are explained in Fig. 1. Labels correspond to mapIDs for each site; Grey numbers:**
**sites used in the development of the metamodels (mapIDs 1-76); Pink numbers: sites without LPM data, used for predictions (mapIDs 77-82).**

We note that the metamodel training (and test) datasets are derived from model-based estimation of age distributions (i.e. from the LPM). This abstraction is necessary where the "truth" cannot be known (i.e. it must be estimated or modelled). As such, the results, and subsequent discussion, presented here are not directed towards assessing the metamodels' abilities to predict
true age distributions (as these are unknowable). Rather, we focus on the ability of these two metamodelling approaches to extract information contained within geochemical datasets for making predictions of groundwater age that are equivalent to those derived from the LPMs.

In this context, the results are summarised in Supplementary Table S2 and illustrated in modelled age distribution plots in Supplementary figures S1-S4. Fig. 6 highlights age distributions for selected sites and demonstrates the variation in model performance between sites and across age percentiles. Please note that in Fig. 6 (and Supplementary figures S1-S4), we have used the median absolute deviation (MAD) of the ensemble instead of the standard deviation, to accommodate the complexity of the metamodel ensemble distributions at each age percentile (refer to Supplementary Fig. S5 for further discussion of this issue). The ensemble MAD is a description of the variation within the model ensembles. As the metamodels are trained at each percentile, the MAD can vary across the predicted age distributions. Even for the chained models, the MAD at a specific percentile need not have a strong relation to the MAD at an adjacent percentile. As discussed in more detail below, we note some general patterns in the results, across sites and age percentiles, for the SR and GBR approaches.

Firstly, the machine learning models generally provide good correspondence to the LPM age distributions. The median ±MAD for the 40-model ensembles generally encompass the LPM derived age, at each percentile (Supplementary figures S1-S4; e.g. "1940_75", Fig. 6). Using $R^2$, median absolute error (MAE) and median relative error (MRE) as correspondence metrics, both approaches appear to perform well, with ensemble mean $R^2$ values generally greater than 0.7 and MRE generally below 10% for "test" datasets, across all nine percentiles (see Supplementary Table S2). For the unchained models, the average (mean) $R^2$ and MAE, across all nine percentiles in the age distribution, were 0.83 and 7.5 years, respectively, for the SR models, and 0.98 and 1.16 years, respectively, for the GBR algorithm. The chaining procedure provided apparent improved correspondence to the LPM for the SR approach ($R^2$ of 0.94 and MAE of 4.4 years). However, chaining marginally reduced the correspondence for the GBR algorithm ($R^2$ of 0.95 and MAE of 2.16 years). The level of correspondence to LPM derived groundwater age estimates indicates that the hydrochemical dataset is a capable estimator of age distributions, equivalent to those provided by the LPM, in this study area.

Secondly, we observe some variation in performance between age percentiles (Supplementary Table S2). Generally, this is typified by greater uncertainty and poorer correspondence at the highest age percentiles, representing the oldest water component at a site (e.g. "Hospital_72" and "Brookvale 1_60" in Figure 6). However, as Supplementary Table S2 shows, the lowest percentiles also, collectively, exhibit a tendency for poorer correspondence. For the unchained models, the ensemble mean $R^2$ values for the SR and GBR algorithms are highest for the 50th percentile and decrease slightly towards both the lowest and highest percentiles. The chained models also displayed this relationship between correspondence and percentile being modelled, though to a less pronounced degree. This may result from the pragmatic censoring of chemistry data at analytical detection thresholds, with the youngest and oldest age fractions being the most likely to have censored hydrochemical results for certain parameters. For example, as discussed in Sect. 2.3, young groundwaters are more likely to be oxic and hence contain near- or below-detection concentrations of Fe, Mn and $NH_3$-N, whereas older groundwaters are more likely to be anoxic and therefore contain near- or below-detection concentrations of DO and $NO_3$-N (Daughney et al., 2010; Morgenstern and Daughney, 2012). Thus, the approach taken in this study of replacing all censored concentrations with their corresponding analytical detection limits may have impacted the ability of the metamodelling methods to discriminate or simulate the lowest and highest percentiles in the age distributions. Overall, the slightly poorer correspondence produced by both the SR and GBR algorithms at the extremes of the age distribution suggests that caution should be exercised when using hydrochemistry-age relationships to evaluate the potential for the presence of contaminants such as pathogens, which tend to occur in the youngest age fraction of a water sample, or geogenic substances such as Fe or Mn, which are more likely to occur in older age fractions.

Third, notwithstanding the general good performance of the two metamodelling approaches, relative to the LPM-derived groundwater age estimates, the correspondence of metamodel age distribution predictions to LPM-derived age distributions does vary between different sites, with better correspondence at some sites than others (e.g. compare "1940_75" and "T2_34"

in Figure 6). As noted above, in the absence of the known truth (which cannot be provided by the reference LPM data), it is impossible to definitively attribute poor correspondence to failure of the metamodels. It does however reflect a break in the complex relationship between chemistry and the LPM estimated groundwater ages, that is established for other sites (e.g. those with good correspondence). This could be the result of local chemical interactions, or characteristics, that are not well captured in the broader dataset or extracted by the models. Alternatively, it could be an indication of complexity in the true age distribution, that is captured in the chemistry but is not well represented in the LPM-derived estimate, based on tracer data.

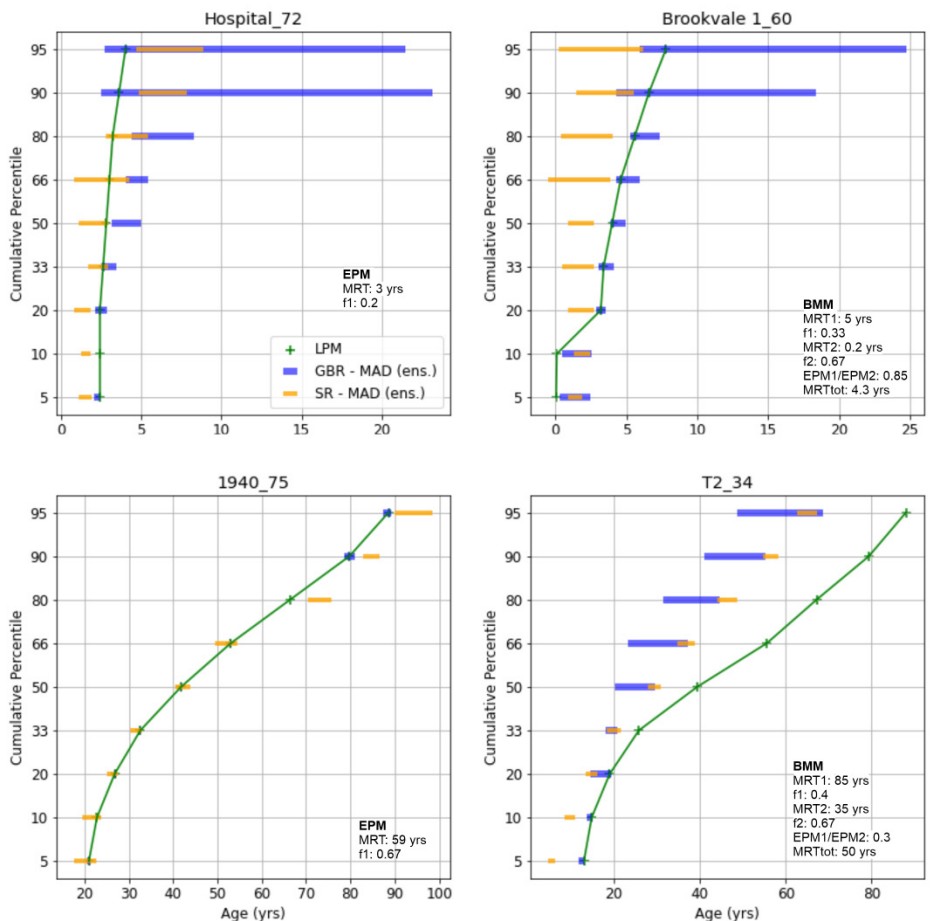

**Figure 6: Age distributions for selected samples based on LPMs (Morgenstern et al. 2018) compared to the chained SR and GBR models developed in this study (bars represent ensemble median ±MAD – see Supplementary Fig. S5 for further discussion of this). Note, the scale of the x-axis (Age) varies between subplots. The legend in the upper left sub-figure is representative for all sub-figures. The mapID following the underscore links with the location of the site on Fig. 3. Age distribution parameters shown are: MRT (MRT1, MRT2) and the fraction (f1, f2) of exponential to total flow volume for each EPM; EPM1/EMP2 is the ratio between two EPMs in a BMM; and MRTtot is the MRT between the two EPMs in a BMM.**

Finally, we note that, at some sites, there is variation in the performance (correspondence to the LPM-derived ages) between the two metamodelling approaches, as illustrated by the variation in MAE illustrated in Fig. 7. The SR method appears to show more spatial clustering in the performance variation. The SR model also appears to show some relation between sample (bore) depth and the model performance; that is, lower correspondence to LPM ages (MAE > 7 year) was confined to samples from bore depths < 50m. Aspects of spatial and depth variation in model performance could be inferred to arise from spatial variations in hydrochemistry caused by groundwater-surface water interaction and groundwater flow paths through the aquifer. The significance of the apparent relationships is hard to distinguish with the relatively small dataset available here and such systematic variations were not as evident for GBR predictions. However, the variations in correspondence to the LPM were not systematically related to the site cluster assignments shown in Fig. 1. This provides some justification for our approach of developing metamodels for all hydrochemical clusters simultaneously (Sect. 3.2).

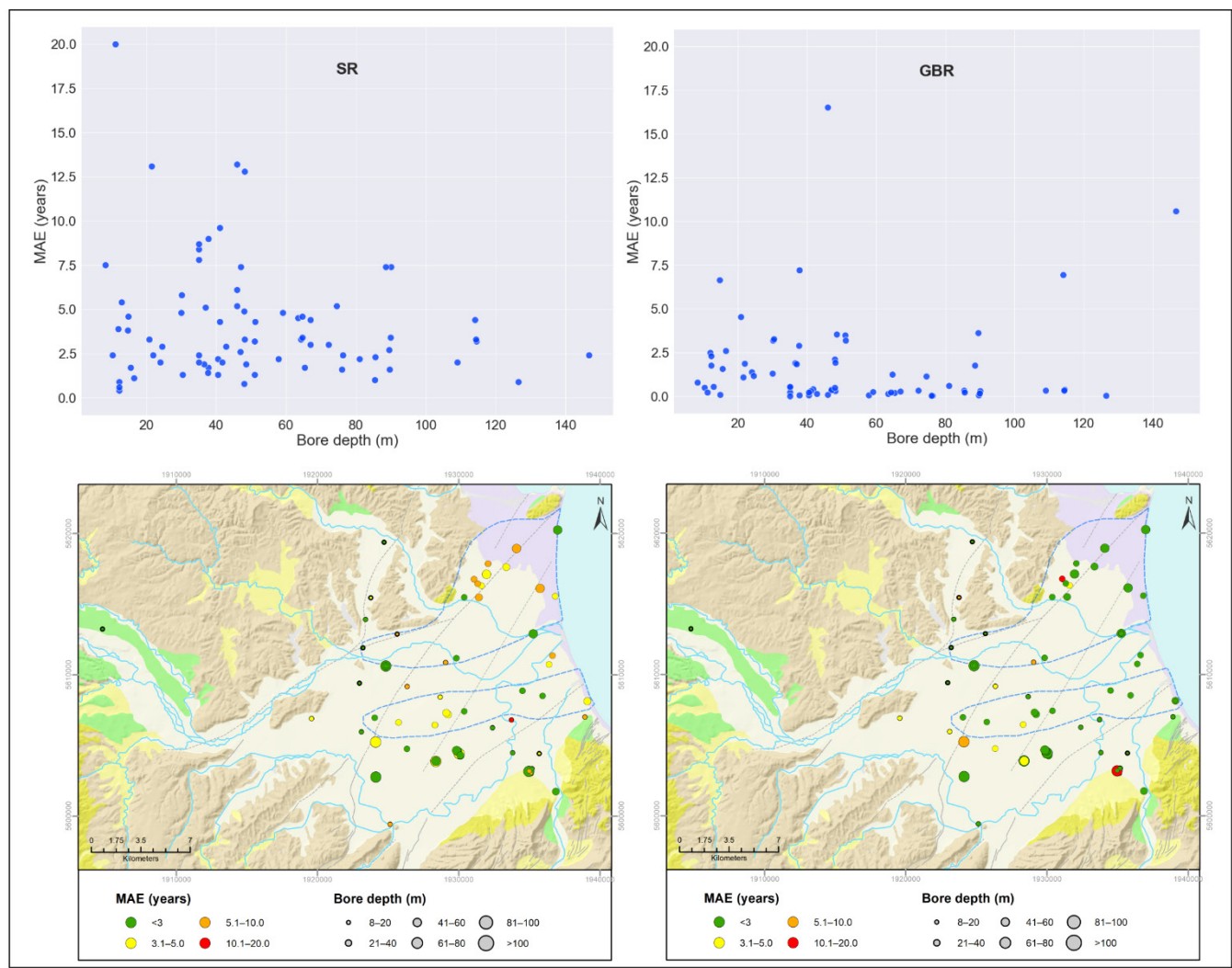

**Figure 7: Ensemble average MAE for each site across all nine modelled percentiles vs. bore depth (top graphs) and site location (bottom figures) for chained SR models (left figures) and GBR models (right figures). On the bottom maps, colours represent MAE and symbol size is scaled to bore depth.**

Based on these results, the SR and GBR methods are seen to produce equivalently good correspondence to LPM derived ages for the application in this study. This is in agreement with studies that successfully predicted groundwater age from hydrophysical data using metamodels (Fienen et al., 2016, 2018; Starn and Belitz, 2018; Starn et al., 2021), in particular with Fienen et al. (2016), who produced comparable results predicting groundwater age using three different machine learning approaches with a consistent input dataset. Both, SR and GBR have their advantages and disadvantages regarding model construction, transparency, adaptability to new parameters or applications, etc. For example, SR provides an explicit model equation as output, which is more transparent and straightforward to apply in other applications but may be subject to injection of bias by the modeller's decisions for the allowable SR grammar rules. GBR is less transparent but may be less subject to the injection of modeller biases. Both SR and GBR can be adapted to new parameters and modelling settings rapidly. The performance of the two metamodelling methods is very similar in terms of providing groundwater age distributions that correspond to those generated from the LPM.

## 4.2. Relationships between Hydrochemistry and Groundwater Age Distributions

Both metamodelling approaches provide an opportunity to explore, interrogate and quantify the respective influence of different variables (e.g. hydrochemical analytes) on the predictions (e.g. age distributions). These provide insights into the relative importance ('feature importance') of recording specific analytes for making predictions of groundwater age and also provides insights into the hydrogeological system itself and the controls on physical (flow) and chemical processes (reactions). The results of this analysis and a discussion on potential insights and implications follows.

The hydrochemical parameters with most influence on the SR and GBR models were identified by scaling the relative variable weights for each model from 0 to 1, then determining the median and MAD of these weightings, within each 40-member ensemble, at each of the five modelled percentiles in the age distribution (Fig. 8). Additionally, for the SR model, a sensitivity analysis was undertaken whereby the value for a given hydrochemical variable was increased or decreased by 10% while the values for all other variables were held constant.

PO$_4$-P (DRP), NH$_3$-N, DO and T (groundwater temperature) were found to be the parameters with greatest overall influence on the unchained models, having median weights across all age percentiles of 0.74, 0.55, 0.46 and 0.48 respectively, for the SR, and 0.37, 0.98, 0.17 and 0.20 for the GBR, respectively. All other hydrochemical parameters had median weights of less than 0.4 and most had median weights of less than 0.2 for the SR. Median weights for all other hydrochemical parameters for the GBR were below 0.2 (Fig. 8). Sensitivity analysis with the SR model showed that, when the values of all other hydrochemical variables were held constant, a 10% increase in PO$_4$-P or NH$_3$-N resulted in a median increase in the estimated age of 3% and 1%, respectively, across all simulated age percentiles, whereas a 10% increase in DO caused a 1% decrease in the simulated age, but only for the 5$^{th}$ to 20$^{th}$ percentiles. Temperature was the most sensitive parameter, with a 10% increase in T causing the estimated age to decrease by approximately 20% across all percentiles. Sensitivity analysis showed that, for all other variables, a 10% increase in value resulted in a change of less than 1% in the estimated age for any percentile. This indicates, that for the specific case of this study area, reasonable estimates of site-specific age distributions can be generated with fewer hydrochemical parameters as input into the metamodels, though this could not have been known *a priori*.

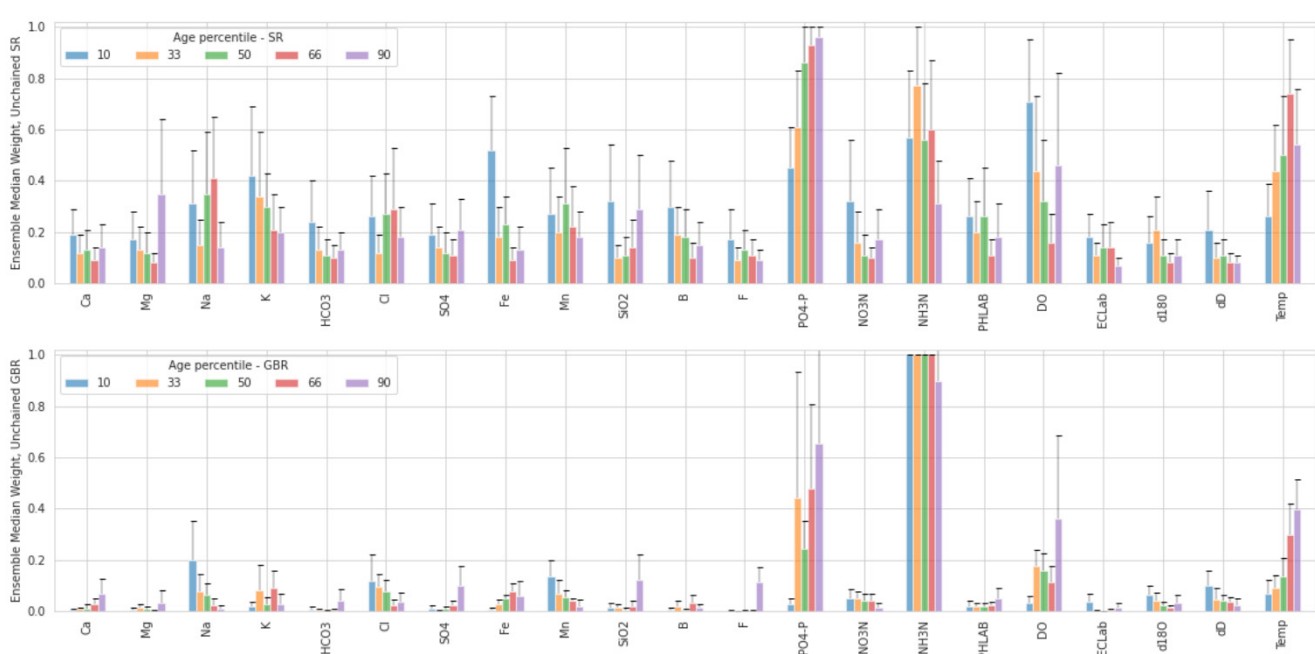

**Figure 8: Ensemble median variable weights (scaled from 0 to 1) from unchained SR and GBR models, for selected percentiles in the age distribution (10th, 33rd, 50th in red, 66th and 90th, shown from left to right for each variable). Error bars represent the ensemble median absolute deviation.**

The importance of most hydrochemical parameters varied substantially within the individual models of each 40-member ensemble, as shown by relative MADs (MAD/median) of up to 60% (Fig. 8). This shows that, for a given age percentile, a particular hydrochemical variable could have high weighting in some models but low weightings for other models. This result likely reflects the high correlation among some hydrochemical variables (Fig. 2). Due to this correlation, if a particular variable is randomly selected for inclusion in the initiation of the SR or GBR algorithm, those variables to which it is correlated provide relatively little improvement in model predictions so tend to be excluded. For example, a model that includes K would be relatively unlikely to include Na, due to the correlation between them. However, because the SR and GBR algorithms are seeded randomly, models that include K (but likely not Na) as well as models that include Na (but likely not K) can be produced within a single 40-member ensemble – resulting in an overall higher MAD and lower median weighting for such variables

across the ensemble. This interpretation is supported by the fact that the variables such as $PO_4$-P, $NH_3$-N, DO and T have relatively low correlations to other variables (Fig 2).

### 4.2.1. Organic matter oxidation

Consistent with a decrease in redox potential over time, the SR and GBR models identify that the concentrations of $NH_3$-N, Fe and Mn all tend to increase with groundwater age, whereas concentrations of DO and $NO_3$-N tend to decrease. These patterns are anticipated, based on observations of other New Zealand groundwater systems (Daughney et al., 2012; Morgenstern and Daughney, 2012) and the known sequence of energetics in the oxidation of organic matter in aquifers (McMahon and Chapelle, 2008). The strong positive weightings of $PO_4$-P in the age models (Fig. 8) are inferred to reflect its release into solution concomitant with reductive dissolution of iron and/or manganese oxide minerals (e.g. Hongve, 1997; Johnson and Loeppert, 2006). Of note, $PO_4$-P is retained as a predictor variable in over 90% of the individual SR models developed across all percentiles, indicating that the geochemical processes that control its concentration are omnipresent across the study area. The SR and GBR models do not identify a strong negative correlation between $SO_4$ concentration and groundwater age (Fig. 8), suggesting that redox potential has not declined to sulphate-reducing conditions at a sufficient number of sites for this relationship to be prevalent in the dataset. Likewise, the SR and GBR models detect only weak relationships between groundwater age and $HCO_3$ or pH, suggesting that the concentrations of these variables are not exclusively controlled by previously reported relationships between organic matter oxidation, alkalinity and acidity (Scott and Morgan, 1990; Sverdrup et al., 2019).

The results of this study can only at best provide semi-quantitative insights into the geochemical kinetics of organic matter oxidation in the Heretaunga Plains aquifer system, based on the rate of decline in redox potential as indicated by changing concentrations of the above-listed redox-sensitive parameters. Evaluation of the concentrations of DO, $NO_3$-N, Fe and Mn indicates that the oxidative capacity of the Heretaunga Plains groundwater is dominated by DO (cf. Scott and Morgan, 1990). As noted above, the models in this study indicate that a 10% increase in DO concentration corresponds to a median decrease of approximately 1% in the estimated groundwater age. Noting that different types of organic matter may oxidise at different rates (Westrich and Berner, 1984), for simplicity we assume that all organic matter in the study area is equally reactive (Middelburg, 1989) and that its oxidation is described by first-order kinetics (Tarutis Jr, 1993), such as $\ln[DO_t] = (\ln[DO_{t=0}] - kt)$ (Langmuir, 1997). As an indicative result only, making the assumption that initial DO and organic matter concentrations were 8 mg/L and within the range 3-8 mg/L, respectively, across all sites, the models developed in this study indicate an average rate constant of $\log k$ (1/y) = -0.6, which is comparable to the values reported by Westrich and Berner (1984) and Middelburg (1989), albeit for marine sediments instead of for aquifers. Greater insight into the rate of organic matter oxidation in the Heretaunga Plains could be gained if future studies make measurements of the concentrations and reactivities of dissolved and solid-phase organic matter.

### 4.2.2. Water-rock interaction

Aside from reductive dissolution of iron and manganese oxides discussed in the previous section, water-rock interaction is expected to increase the dissolved concentrations of mineral-forming elements such as Ca, Mg, Na, K or $SiO_2$ (Sverdrup et al., 2019). These five hydrochemical parameters have low median weightings in the SR and GBR models developed in this study (Fig. 8), suggesting that they are not important predictors of groundwater age at the majority of sites. However, these same five parameters were retained in close to half of all models across all percentiles, suggesting that they are important predictors of age for at least some sites.

We infer that the concentrations of Ca, Mg, Na, K and $SiO_2$ are retained in the SR and GBR models primarily as a means of differentiating rainfall-recharged groundwaters from river-recharged groundwaters, which then allows the algorithms to apply appropriate age estimations depending on the relevant recharge source. As noted in Sect. 2.3, river-recharged groundwaters

typically have slightly lower concentrations of Ca, Mg, Na, K and/or $SiO_2$ compared to rainfall-recharged groundwaters, which results from the relatively faster accumulation of these substances during the passage of water through the soil zone, likely due to mineral dissolution (e.g. of carbonates), ion exchange and evaporation (Morgenstern et al., 2018). This inference is supported by the fact that the age estimates for the subset of rainfall-recharged sites are generally more sensitive to the concentrations of Ca, Mg, Na, K and $SiO_2$ used as input to the models. This inference would be usefully tested through future investigations that evaluate the respective roles of silicate mineral weathering, ion exchange and evaporation during the passage of recharge through the soil zone in the Heretaunga Plains.

### 4.2.3. Human impacts

The SR and GBR models do not identify strong relationships between groundwater age and any of the commonly analysed indicators of human impact on groundwater quality. In New Zealand, human impacts on groundwater quality are most readily identified by elevated concentrations of $NO_3$-N, sometimes co-occurring with elevated concentrations of Na, K, Mg and/or Cl (Daughney et al., 2012; Morgenstern and Daughney, 2012). That $NO_3$-N is not a strong predictor of groundwater age in the Heretaunga Plains likely reflects that many of the sites are recharged from rivers, which have lower $NO_3$-N concentrations compared to groundwaters that are recharged from rainfall (Sect. 2.2), and/or that the degree of impact evident in the recharge water has not changed substantially over time. Elevated concentrations of $PO_4$-P in New Zealand groundwater can arise from dairying land use, especially over gravel or sand aquifers (McDowell et al., 2015), but such land use is not common in the Heretaunga Plains (Smith et al., 2020), and hence $PO_4$-P concentrations are instead inferred to reflect geogenic origin (Sect. 4.2.1). Concentrations of pesticides, emerging contaminants or microbial pathogens can also reveal human impact on groundwater quality but were not analysed in this study.

### 4.2.4. Temperature

The SR and GBR models reveal a strong inverse relationship between T and estimated groundwater age, i.e. as T increases, the modelled groundwater age decreases. Particularly for the higher age percentiles, T is among the variables with the highest median weightings across the model ensembles (Fig. 8). The available data from this study do not permit elucidation of the cause(s) of the strong inverse relationship between T and estimated groundwater age, but the following paragraphs present two concepts that could be explored through further investigations.

One possibility is that an increase in temperature causes an increase in geochemical reaction rates, such that groundwaters interpreted to be younger based on the models developed in this study are also seen to be warmer. The relationship between T, reaction rates and estimated groundwater age can be semi-quantitatively evaluated using the Arrhenius expression (Eq. 4):

$$\frac{d \log k}{dT} = \frac{E_a}{2.303RT^2} \tag{4}$$

where k is the rate constant, $E_a$ is the activation energy, R is the gas constant, and T is expressed in the Kelvin scale. For the comparison of reaction rates $k_1$ and $k_2$ at two temperatures $T_1$ and $T_2$, the above expression can be arranged (Langmuir, 1997) as Eq. (5):

$$\log \frac{k_1}{k_2} = \frac{E_a}{2.303R}\left[\frac{1}{T_2} - \frac{1}{T_1}\right] \tag{5}$$

In the application to the present study the above equation is aggregated across all reaction types because there is no available means of identifying specific types of reactions that may be more important than others. On this basis, the above equation suggests that the average $E_a$, aggregated across all reaction types, is approximately 25 kcal/mol, derived from $T_1 = 15°C = 288K$ (the median across all samples), $T_2 = 16.5°C$ (a 10% increase above the median) $= 289.5K$, and $k_1/k_2 = 0.8$ (because the models developed in this study indicate that a 10% increase in T causes a median decrease of approximately 20% in the estimated age). This estimated value for $E_a$ is in the range expected for mineral dissolution reactions (8-36 kcal/mol) and ion exchange (>20 kcal/mol) (Lasaga, 2018) and for organic matter decomposition (ca. 20-30 kcal/mol) (Leifeld and von Lützow,

2014). The correspondence between these previously published values of $E_a$ and the estimate derived in this study suggests that the effect of T on modelled groundwater age may indeed be driven by increases in the rates of reactions such as organic matter oxidation and water-rock interaction as discussed above. Accordingly, we surmise that T is retained in the models as an important modifier of the effects of such reactions on hydrochemistry. However, further research is required to rigorously test this possibility.

Another possibility is that T may affect the estimated groundwater ages through hydrologic factors, rather than through geochemical kinetics as described above. For example, one possibility is that there is a significant difference in the temperature of slower-moving groundwaters that are recharged from rainfall compared to faster-moving groundwaters that are recharged primarily from river seepage (see Fig. 1). This hypothesis is not supported by the measured values of T, which show no significant differences arising from the inferred groundwater recharge source. Moreover, it is probable that river-recharged groundwaters, being sourced from higher altitude precipitation, would be cooler than rainfall-recharged groundwaters, which would lead to a relationship between temperature and age that is opposite to observed in this study. However, this study is limited by a relatively small number of samples, so further investigation with collection of samples across a wider range of seasons and recharge conditions would be beneficial to elucidate any hydrological controls on the observed relationships between T and groundwater age.

### 4.3. Applications

#### 4.3.1. Estimation of groundwater age distributions without age tracer data

The SR and GBR models developed in this study can be used to estimate the values for the nine specific percentiles in the age distributions solely on the basis of groundwater chemistry, i.e. for sites and samples for which age tracer data are not available. To illustrate this application, we make use of data collected through the Hawke's Bay Regional Council groundwater quality monitoring programme (Supplementary Material Table S4). These samples are collected using the same protocols and analysed with very similar procedures for the same variables as described in Sect. 3.1, so are considered suitable for use with the SR and GBR models developed in this study. The only exception is for B, F, $\delta^2$H and $\delta^{18}$O, which are not routinely measured by the regional council; however, these four variables all have low influence in the SR and GBR models (Fig. 8), and so we applied average values derived from all other sites, which we conclude would have had little influence for this application.

The spatial variations in groundwater age for sites without available age tracer data are shown in (Fig. 5). In some areas, the SR and GBR estimates provide infilling of modelled age in between the locations where LPMs are currently available. For example, the SR and GBR age estimates can improve the understanding of the demarcation between the zones of younger river-recharged groundwater in contrast to older rainfall-recharged groundwaters (Fig. 1). The SR and GBR models can also estimate groundwater age quite distant from the nearest sites having available LPMs, thereby providing useful information for groundwater management where such information was previously lacking.

The temporal variations in groundwater age can also be assessed using the SR and GBR models at sites for which time series groundwater chemistry data are available (Fig. 9). Application in this study suggests that the temporal shifts in the groundwater age distribution at a single site can be substantially larger than the ensemble MAD for a single sampling date. This suggests that temporal or seasonal variations in the groundwater age distribution can be reasonably large at some sites. This inference is supported by the age tracer results for the few sites that had been sampled on more than one occasion. For these sites, there were cases where the LPM age distributions were inferred to vary temporally or seasonally, based on observed shifts in the concentrations of the age tracers; these sites also displayed temporally variable hydrochemistry. Thus, it is reasonable to anticipate that seasonal or longer-term variations in recharge and/or abstraction on groundwater flows that are known to affect groundwater age distributions (Engdahl et al., 2016; Toews et al., 2016; Yang et al., 2018) may be indicated by temporal shifts in groundwater chemistry. The SR and GBR models developed in this study indicate such shifts in groundwater age distribution. Moreover, different percentiles in the age distribution at a single site can display quite different temporal patterns,

as shown by a relatively constant 50<sup>th</sup> percentile but more variable 10<sup>th</sup> percentile at Site 413 (mapID 20), or the opposite
pattern at Site 611 (mapID 77) (Fig. 9).

We acknowledge that the SR and GBR models developed in this study were based primarily on samples collected in the period April to June and the years 2014-2019 (Sect. 3.1), hence caution must be exercised for their application to other seasons or time periods. We also acknowledge that the metamodels described in this paper are specific to the training region. While the same metamodelling approaches may be used elsewhere where there is sufficient groundwater chemistry data, the same

metamodel hyperparameters are unlikely to apply in other regions (Doherty and Moore, 2021). Therefore, age tracer training data sets would be required also for other regions. However, within a single hydrogeological setting, the metamodelling approach enables a space-for-time substitution while preserving the key processes that relate groundwater chemistry to groundwater age. We therefore conclude that the SR and GBR models can offer useful insights to spatial and temporal patterns in groundwater age distribution based on chemistry and therefore assist sustainable groundwater management if age tracer data

are not available.

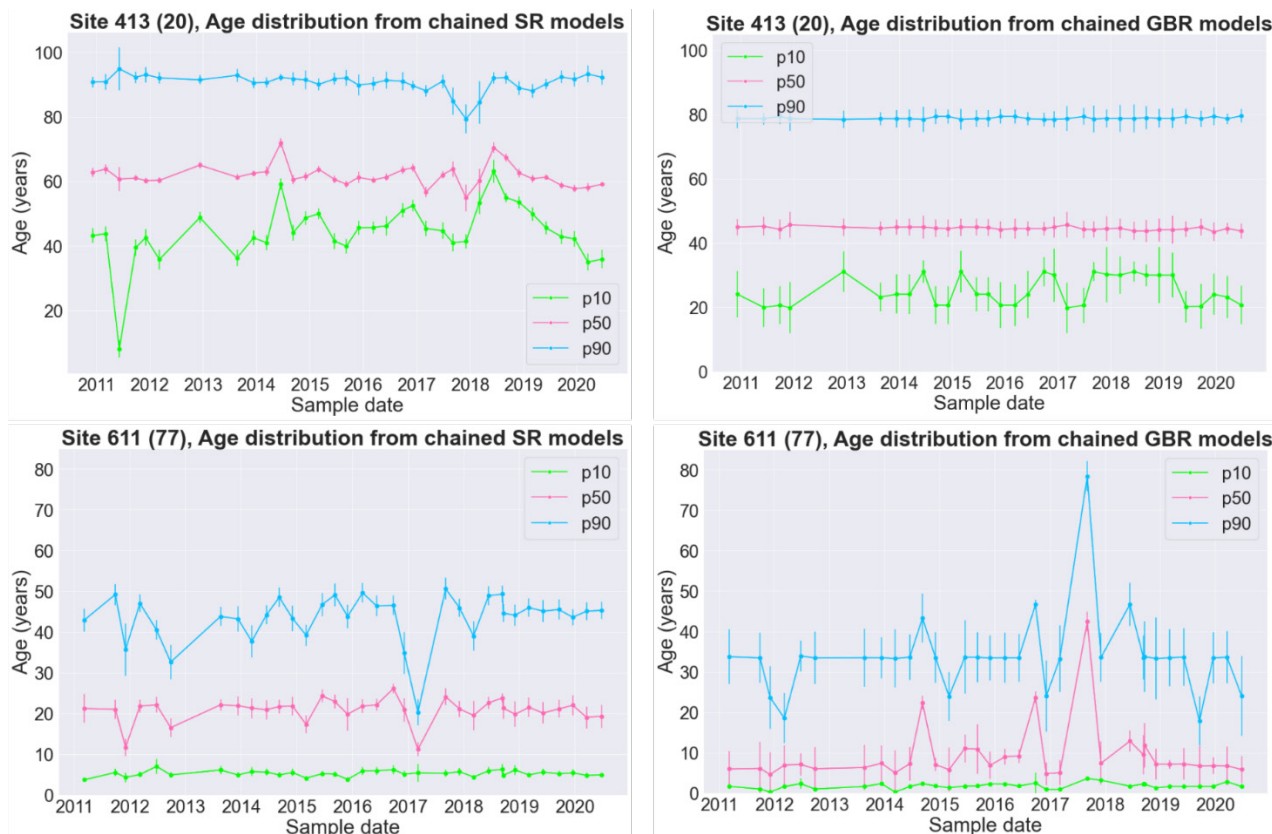

**Figure 9: Temporal patterns in the estimated values for selected age percentiles at two sites based on SR models (left) and GBR models (right). Points and error bars represent the median and MAD for the relevant 40-member model ensemble, respectively. MapIDs in brackets link to the location of the sites on the map in Figure 5.**

**4.3.2.    Constraining LPMs**

The approach taken in this study is to treat the LPM-based age distribution as reference, which the SR and GBR models are subsequently developed to reproduce. The SR and GBR models are therefore acting to generalise the relationships between chemistry and the LPM age percentiles based on a set of independent samples from a range of sites. That the SR and GBR models perform well across the Heretaunga Plains dataset indicates that there are generalisable relationships between

groundwater chemistry and LPM age. Thus, samples for which the SR and GBR models perform poorly may indicate that the LPM age distribution is in fact in error and could be better constrained if outputs from the SR and GBR models were taken into account.

The SR and GBR models may assist the choice of mixing model to be used when fitting an LPM to age tracer data. In the absence of any other information, the general practice is to select the simplest LPM mixing model that provides an adequate

fit to the available age tracer data, in accordance with the principle of parsimony. Thus, an EPM (having just two optimisable parameters) is typically the mixing model applied for sites that have been sampled for age tracers on only one occasion, because a more complex BMM (with five optimisable parameters) would be under-constrained and therefore unjustifiably complex. However, the age distribution inferred from groundwater chemistry may indicate sites for which a BMM is more appropriate, even if the available age tracer data can be adequately fitted by an EPM. This case is illustrated for Site 1940 (mapID 75), which has only been sampled for age tracers on one occasion, but for which the SR and GBR models imply a more complex age distribution more consistent with a BMM than an EPM (Fig. 6). While we recommend that the fitting of the LPM should be based on the age tracer data, we suggest that the SR and GBR models can usefully guide which LPM mixing model(s) may be appropriate.

The SR and GBR models may also help to constrain the LPM parameter values, in particular in the Northern Hemisphere, where tritium age interpretations are still ambiguous. Depending on groundwater age, and time and frequency of age tracer sampling, for some sites, the age distribution parameters can robustly be constrained. These could be used for training the metamodels which could then be used to help estimate the parameters for the sites where age tracers are insufficient to constrain all model parameters. Specific applications could also be to use the metamodels to constrain the second LPM model parameter where only one age tracer measurement in time is available, and to demonstrate variable ages in a well where lots of hydrochemistry data is available (e.g. quarterly monitoring) but only few age tracer samples.

### 4.3.3. Informing groundwater model-based management decisions

The metamodels developed in this study are able to enhance our understanding of groundwater age distributions across an aquifer system, through extrapolation of age tracer data to any site where there is groundwater chemistry data. This can then provide greater insight into how the groundwater systems functions, in particular groundwater recharge rates, the location of aquifer discharge areas, and the provenance of recharge sources. This information can also inform estimates of groundwater travel times, how these travel times they may vary across an aquifer system, and how this may affect well source protection zone delineation. Estimates of age information can inform us of how an aquifer system may be changing over time in response to changing climate and/or abstraction pressures. In all of these ways, this information helps inform a conceptualisation of a groundwater system required for the construction of numerical models used to support groundwater management decisions (Ferguson et al., 2020).

The estimated ages derived from the metamodels can also be used provide history matching targets for numerical models that are used to inform, and reduce the uncertainty associated with, groundwater management decisions (Sanford 2011, Koh et al 2018). Wilcox et al. (2021) discuss various opportunities offered by such combinations of physics-based modelling with metamodeling, to bridge the respective limitations of data-sparsity, and the need for physics-based constraints to ensure reliability and viability.

### 5. Conclusions and Future Work

Overall, this study has shown that SR and GBR are useful approaches for codifying the relationships between hydrochemistry and groundwater age at the aquifer scale. This finding is consistent with previous studies that have identified statistical or first-principles relationships between groundwater chemistry and age (e.g. Daughney et al., 2012; Morgenstern et al., 2012; Beyer et al., 2016, Sverdrup et al., 2019). The key advance in this study is to extend these hydrochemistry-age relationships to specific percentiles in the age distribution, thereby providing greater insights for groundwater management, such as the potential for occurrence of young or old groundwater fractions that may be associated with specific types of contaminants. Both metamodel approaches are shown to effectively estimate groundwater age distributions that correspond with LPM age distributions, from hydrochemical analytes within the hydrogeological context of the training region. As well as providing predictions of age

distributions, which can directly inform system understanding and management, the analysis presented here also provides insight into the chemical pathways that are active in the study region.

We identify three avenues for extension of this study.

Firstly, we note that the metamodels generated in this study are likely to be specific to the study region. The SR and GBR approaches could be extended to estimate groundwater age distributions from hydrochemistry in other aquifers, or even at the national and international scales. Such work would serve to identify the universality of, or limits to, transferability of the age-hydrochemistry relationships and the models that encode them. A related opportunity is to investigate the utility of other datasets alongside hydrochemistry, e.g. well location, depth, elevation, etc., which may improve the elucidation of age-chemistry relationships identified in this study. A second opportunity for further work is to apply the age-hydrochemistry relationships to improve or calibrate kinetics models of water-rock interaction or biogeochemical processes (e.g. Sverdrup et al., 2019).

A third opportunity for further work is to apply the age-hydrochemistry relationships in model-based groundwater management decision support. Estimated groundwater ages in combination with other geological information can inform groundwater system conceptualisations, such as the spatial disposition and rates of groundwater recharge and discharge. Groundwater age estimates can also be used as history matching constraints to inform and reduce the uncertainty of groundwater related risk assessments (Wilcox et al., 2021, Sandford, 2011). These could include for example, the security of groundwater supplies in a changing climate, the occurrence and transport of contaminants, and the moderation of groundwater abstraction regimes.

**Data availability**

The datasets relevant to this study are available from the corresponding author on request.

**Author contribution**

Conceptualisation and funding acquisition: CM, UM, CD. Supervision: BH, CM. Data curation: CT, CD, SK. Investigation & methodology: CD, SK, CT, BH. Writing and visualisations original draft: CT, CD, SK. Writing review & editing: CT, CD, SK, BH, CM, UM

**Competing interests**

At the time of writing, Chris Daughney held a half-time secondment as Chief Science Advisor for Te Uru Kahika Regional and Unitary Councils Aotearoa. The other authors declare that they have no competing interests.

**Acknowledgements**

This paper represents a collaboration between NIWA and GNS Science, in a GNS-lead programme called Te Whakaheke o te Wai, funded by the New Zealand Ministry of Business Innovation and Employment (MBIE, grant number C05X1803). Co-funding was provided by GNS′ Groundwater research programme (MBIE, Strategic Science Investment Fund, grant number C05X1702). The authors would also like to thank Hawke's Bay Regional Council, and Hastings and Napier district councils for providing data. Thanks also go to the Te Whakaheke o te Wai project partners and Science Advisory Panel for advice and support. Additional thanks goes to the reviewers and the journal editor for their insightful and constructive comments.

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
