# Peer review of "Estimation of groundwater age distributions from hydrochemistry: Comparison of two metamodelling algorithms in the Heretaunga Plains aquifer system, New Zealand"

_Hydrology and Earth System Sciences, 2022_

## Author Comment (AC3)

We thank all four reviewers for their insightful and constructive comments, shown in black text below, to which we respond below using blue text.

We have addressed all concerns and will action most suggestions in a revised manuscript, as detailed in our responses. As per the instructions of the journal, a revised manuscript has not been submitted at this stage.

We have combined the review comments, and our responses, from all reviewers into one document and introduced numbered subheadings in the remainder of this document so that we can more easily cross reference our responses between the different reviewers who raised similar concerns. For each reviewer, we have identified the comments that we consider to be more substantive versus those that are relatively minor.

In our view, the most substantive comments from the reviewers pertain to:

- Whether the chemistry-based metamodels should have been trained by matching to LPM-derived age distributions as opposed to some other information source (Reviewer RC1, RC2 and RC4)

- Whether the sites should have been segregated into different chemical clusters, then metamodels developed for each cluster separately (Reviewer RC1)

- Our explanations of the drivers for the inferred age-hydrochemistry relationships (Reviewer RC3)

- The appropriateness of the chaining approach we have employed (Reviewer RC4).

**Contents of this document**

**1.1    RC1 – Scott Wilson**

This paper makes an excellent and novel contribution to predicting transit times using a wider dataset than isotopic age tracers. I found the text enjoyable and easy to read, and the perspective is fairly balanced. I have two main comments on the approach taken, which have some bearing on the conclusions that can be derived from this work.

**1.1.1    Training chemistry-based metamodels on LPM-derived age distributions**

The main drawback of the approach taken is that the chemically based models use the lumped model age estimates as a response variable, ie training a model on another model which is acknowledged as having shortcomings, although this is the motivation for the paper. The difficulty is that this creates an ambiguity as to whether mismatches in the trained models are due to poor lumped model estimates, or poor local performance of the trained models, or both, or something else (eg parameter or model selection). As a suggestion, an alternative or complementary approach would be to firstly train models to predict the isotopic tracer con\centration. This would provide some prior information on the mismatch between the lumped model predictions, and ensemble predictions. This approach could perhaps inform how the lumped parameter estimates could be improved, which was suggested in 553.This is not a step necessary for this paper, but perhaps something that could be carried out in future work.

This is a very good comment, which we interpret may have arisen in part because our original manuscript neglected to explicitly make the point that groundwater age and groundwater age distributions cannot be measured directly – *they must be inferred using some form of model*. Our original manuscript also laid out the two main options for modelling groundwater age – LPMs vs. numerical flow/transport models – and discussed their strengths and weaknesses.

- We will change the original manuscript to make this point that groundwater ages must be derived from models.

The reviewer commented that LPM-derived age estimates carry some uncertainty which would then be brought over into the metamodels. We agree, and this is of course also true if the age estimates had been derived from a numerical flow/transport model instead. We will modify the text to:

- emphasise that metamodels will inherit the uncertainties of the models they were trained on.
- make an additional point about potential spatiotemporal correlation and bias in the age estimates from LPMs vs. numerical flow/transport models. The point is that both types of models contain uncertainties, but the LPM-derived ages may be less subject to spatially correlated biases. This can be a strength or weakness depending on the objectives of any given study, but we consider to be helpful for our present investigation, which we will note in the updated version of the manuscript.

We acknowledge but disagree with the reviewer's suggestion that our metamodels could have been trained directly on the measured concentrations of the individual age tracers, rather than on a single LPM-based age model based on a combination of all age tracer data. This is for several reasons. Firstly, the different age tracers have different age ranges over which they are valid. Secondly, some of the age tracers are gases whereas tritium is part of the water

molecule, so are subject to different processes in the aquifer system (e.g. gas exchange). Thirdly, some of the age tracers can be subject to degradation or alteration in the aquifer, so the measured concentrations in the sample may not represent their concentrations in the original groundwater recharge. For all of these reasons, best practice in our lab and elsewhere is to carefully compare and evaluate the time-series measurements from all available age tracers and derive the most robust LPM interpretation consistent with them all. As a side note, we do agree with using the measured concentrations of the age tracers when calibrating a numerical flow/transport model, but that is not directly relevant to the present manuscript.

- We will adjust the original manuscript to explain why the LPMs are developed by fitting to all available age tracer data for the relevant site.

The reviewer is correct though, that we have considered that errors in the underlying LPM interpretation may be one reason why the metamodels fit more poorly at some sites. Indeed, a whole section of the original manuscript was dedicated to this idea (Section 4.3.2).

- No modification of Section 4.3.2 is expected, because we consider that it already encapsulated the point being raised here by the reviewer.

**1.1.2 Segregating sites based on chemistry prior to developing metamodels**

The modelling approach applied here is to generate a global model from a subset of individual models, and it is assumed that the input data are spatially and temporally independent. However, the hydrochemical clustering results do indicate that the chemistry data have a predictable spatial and temporal variability. Some evidence of this influence is apparent in the results (eg Fig 6, T2_34), and hence in the applications section some spatial and temporal discrepancies are acknowledged. To overcome this, it may have been beneficial to train models on each hydrochemical cluster, although it has to be acknowledged that there is little data available for clusters 4 and 5. An alternative approach would be to introduce some additional predictive parameters in the model to account for spatial and temporal variability, eg elevation, depth, position, hydrochemical cluster. Some of these parameters have been used for validation, but they could also have been training parameters, or tested to see if they do inform model predictions. In doing so, one could have more confidence in the application of the models to areas with no age data.

We agree that in some applications it can be helpful to pre-segregate observations and develop metamodels on the populations separately. Indeed, we had considered this approach as we were developing our own methodology. However, we opted to attempt to develop a single metamodel for the entire dataset, without pre-segregation, primarily to determine whether the SR and GBR algorithms could themselves account for any inherent differences in the age-chemistry relationships between sites. As shown, our results demonstrate that the models did perform well across all hydrochemical clusters.

- At the start of the section in which the SR and GBR methodologies are described, we will insert a short paragraph to explain why we did not use pre-segregation of sites.
- We will add a single sentence in the results section to explain that site-specific biases in model fit were not systematically related to the site's cluster assignments shown in Figure 1, which justifies our approach of developing metamodels for all hydrochemical clusters simultaneously,

We appreciate the reviewer's suggestion that other datasets such as elevation, well depth, well location could also be used as input data, along with the site-specific hydrochemistry data. Indeed, assessing the information content of different input datasets is one of our research interests, and we are currently looking into using hydrophysical parameters in addition to the hydrochemistry. This is planned for a future paper.

- We will add a sentence about future research in this direction to Section 5.

**1.1.3    Minor comments**

The paper would also benefit from some corections and the clarification of some points listed below.

Title and line 60: SR and GBR methods are not metamodels per se. They are applied in this paper as metamodels because they are trained on the LPMs rather than raw observation data

We agree and will change the introduction to clarify how we use the term metamodel in this paper.

Line 97: Should be Heretaunga Plain not Plains (also elsewhere)

No change will be made to the manuscript because "Heretaunga Plains" is the most common usage and is also being used by Hawke's Bay Regional Council, the regional governing authority for this area (e.g., https://www.hbrc.govt.nz/environment/aquifers/, https://www.hbrc.govt.nz/home/article/851/the-plan-for-healthier-heretaunga-plains-waterways?t=featured&s=1 ).

Line 125: There are red lines on Fig 1 which are unreferenced. Are these flow barriers? It seems odd that there is a flow path towards a flow barrier (centre top)

Good find. The red lines identify areas where Morgenstern et al. (2018) found indication that there is no surface water flow contributing to the main aquifers. We will add an explanation to the figure caption.

Line 154: The clustering detailed in the hydrochemistry section provides some background context, but is not used in the modelling or subsequent analysis.

See our reply in subsection 1.1.2 above. As noted, we now explain why the hydrochemical clusters are not used as input to the metamodeling, and we discuss the implication of this approach in our results section.

Line 219: It's good practice to state that this is the response variable for the statistical modelling, and the hydrochemistry data are the predictor variables

Good point. We will amend the text to include this.

Line 247-249: How much error is the distance to these input signal datasets likely to introduce to the age estimates, and how would that compare to the error introduced by the EPM?

We agree that if a different model was used, that spatial correlations within their error functions can be explored, however in this project we adopted an average error term for simplicity. Spatial correlation within error functions could be investigated in future work. However, we believe that this choice doesn't undermine the general conclusions of this work. Generally, the model simplification error introduced by the LPM would tend to create larger errors than the spatial distance to the input signals. As also discussed in Section 1.2.1 in this document, a detailed analysis of the possible errors and their propagation through to the age distribution estimates is a complex topic and beyond the scope of this paper.

Line 269: Were not was

This seems to refer to this sentence: "*SR grammar rules permitted arithmetic, exponential and logarithmic functions; permission of conditionals (e.g. if/then statements) was also assessed in terms of ability to improve model fits.*" We consider that the original grammar was correct because the word 'was' refers to the word 'permission', which is singular.

Line 273: The primary aim of tuning is to improve model performance, not assist convergence

Agree. We will change the text accordingly, as follows: 'The hyperparameters were tuned to find the optimal parameters (tree depth = 4, sample split = 2 and learning rate = 0.05) that result in the best performance of the models.'

Line 278: The terms 'chained' and 'unchained' models is unorthodox, and perhaps not an apt description of what the models represent. Perhaps these would be better referred to as 'independent' (see line 276) or 'individual models', and the chained models as 'ensemble models'

No change has been made in response to this reviewer comment, though we do acknowledge that the terminology is tricky, given the complexity of the methods we have used.

The terms for 'unchained' and 'chained' models are based on the "chained regression" modelling approach, which is for example used in the RegressorChain scikit.learn class that we are employing in the GBR models. This terminology is widely used.

We do not favour the reviewer's suggestion to replace the term 'unchained' with a term such as 'independent' because for clarity we repeatedly use the latter term to emphasise that unchained models are constructed individually (i.e. independently) for each of the modelled percentiles (the chained models are too). To use the term 'independent' in more than one way would make the manuscript more difficult to follow.

We also do not favour referring to the chained models as 'ensemble models' because we in fact applied an ensemble model approach with both the 'unchained' and the 'chained' models. We believe this is already adequately explained in the text and depicted in Figure 4.

Line 286: Why do the train/test splits differ for the two models? This approach doesn't enable a clear comparison of modelling performance between the two models to be made

We compared a range of test/train split ratios based on typical approaches used by practitioners of these modelling methods. We will amend the text to point this out and will

also modify the same paragraph to explain that the final selection of test/train split was based on these tests that we had conducted.

For GBR, we found that a 10/90 % test/train split (i.e. K = 10-folds) achieved a better, more consistent performance with our smaller data set (76 inputs) compared to larger split ratios (e.g. 20/80% and 30/70%). When we decreased K (e.g., K=5 etc.), we provide a smaller training set for GBR models to learn from, and the performance of the GBR models is limited by the amount of data.

For SR, a test/train ratio of 33/66 provided good stability. Due to the random sampling procedure in the SR technique, we understand that even if we had followed the same split ratio, there could be some method specific small discrepancies in the test/train sample selection procedures between SR and GBR procedures.

286: As a comment, a 10/90 split is quite heavy-handed and could lead to overfitting. The unchained GBR R2 values are very high, although this is also true for the SR R2 values

No change has been made in response to this comment. Here, the split (K) is defined by the kfold cross-validation procedure, and this 10-fold cross-validation resulted in 10/90 % test-train split. 10/90 % test-train split in GBR achieved a better, more consistent performance with our small data set (76 inputs) compared to larger split ratios (e.g. 20/80% and 30/70%). Scikit learn cross-validation usually uses 10-fold as the default value and it is known that this 10-fold work well with smaller data sets as the models are trained with 90% of data, resulting in lower bias in estimates. Also, by using a strict model selection criterion we have minimised the effects of overfitting. Also see response to previous comment.

Line 290: There seems to be an error in the Pearson formulas

Good catch. Symbols must have been converted between versions. We will correct this in the updated document.

Line 375: Last Glacial (is a noun)

Good point. Will change this.

Line 399: The third value is 1.7 (ie >1)

Also a good catch. Will correct this with 0.17.

Line 405: Perhaps the models could achieve good age distributions with substantially less parameters?

We agree that our results demonstrate that good estimates of the age distributions can be achieved with fewer chemical parameters as input. While this is implicit in the discussion in our original manuscript, we didn't say so directly so will amend that.

Line 410: It might have been more informative to plot the cluster results here rather than the ensemble weights, since the most informative parameters are already described in the text. As a reader, I'm intrigued by the relationship between the model performance and the clusters.

No additional change has been made in response to this comment. We opted to present the results according to parameter weights in order to demonstrate the hydrochemical variables that exert greatest influence in the model fits across all sites. As per subsection 1.1.2 above, we will already add a sentence to explain that there were no strong differences in model fit for the different clusters, at **Line 378.** Indeed, some of the clusters contain very few sites so we would not likely expect to distinguish statistically different parameter weights if metamodels were developed for clusters individually.

Line 434: Perhaps water chemistry has some influence of the source rock, which wouldn't necessarily be reflected in the age estimates

No change has been made in response to this comment because we consider that it is consistent with what the original manuscript already reported. We agree that the chemistry at any point reflects the water-rock interaction both from the point of recharge and along the whole flow path to the sampling point. There are other factors that would also affect the evolution of chemistry over time, such as microbial processes. As noted in subsection 1.1.2 above, our approach in this study was to attempt to develop metamodels for all sites simultaneously, to determine whether they could identify these age-chemistry relationships themselves. Our methods were shown to be effective in this context, but metamodels would need to be retrained if these same methods are to be applied in a different catchment. Line 517: It's ambiguous how these parameters were treated. Were their values set to the detection limit?

As the parameters B, F, oxygen-18 and deuterium are not routinely monitored, we used dummy values equal to the average across all samples in the predictor dataset. Due to the generally low weighting of these parameters, there should be very limited impact from this on the results. We will clarify this in the text.

Line 522: I think this claim is a bit of a stretch since there are no spatial aspects to this study. The model is aspatial, and global, and appears to generalise well to most, but not all the data. The model has the potential to be applied to other areas with confidence if the successful or unsuccessful predictions could be identified as having an association with something eg a particular cluster. NB this comment also applies to the last sentence of the abstract.

We accept this comment. We will remove the term 'spatial extrapolation', but we maintain that the SR and GBR models can produce estimates of age distribution in areas of similar hydrogeological regimes even where no age tracer measurements have been made. The application described in Section 4.3.1 illustrates that doing so provides rich information about age distributions in areas where they would not have otherwise been available.

Line 543-547: I don't think these statements are valid, particularly in light of the preceding sentences. There is no spatial aspect to the modelling to this modelling approach, it only uses age and chemistry data.

We feel that the reviewer has slightly misinterpreted the intent of statements made in the original manuscript. Our view is that the *metamodeling approaches* demonstrated in this paper can be applied in other catchments where sufficient age and chemistry data are available. While we already note that it is not reasonable to apply a metamodel from one catchment to another catchment, we state that within a single catchment the metamodels can provide estimates of age distribution where no age tracer concentrations have been measured,

as long as chemistry data are available. We will slightly modified the sentence on **Line 543** to indicate that applications within a single catchment must pertain to a single hydrogeological regime.

Line 578: Which of these models would you have the most confidence to apply elsewhere?

The performance of these models in other catchments will likely be comparable if the catchment has a similar hydrogeological regime. The similarity or not of the hydrogeological systems and the key hydrochemical processes are likely going to be the most significant considerations for applying either of these modelling approaches, rather than any differences between these modelling approaches. As mentioned above, the purpose of the paper was not to compare the model performance but rather to demonstrate the utility of metamodels in this context using two different models and their typical model set up (mimicking practitioners use of these modelling methods). As stated in the original manuscript (from line 383 onwards), we believe that either method would be suitable for similar application elsewhere, depending on user requirements, experiences and understanding.

**1.2     RC2 – Camille Bouchez**

This work explores the use of metamodelling techniques to predict groundwater age distributions from hydrochemistry. It is a novel and interesting contribution aiming at increasing the availability of groundwater age information from easily available hydrochemical data in catchments. The knowledge gap is convincing and the paper is nicely written. However, I have some comments that should be addressed before publication.

**1.2.1     Training chemistry-based metamodels on LPM-derived age distributions**

My main concern comes from considering the LPM-derived age distributions as the true representation of groundwater age distribution, which is later used as the metamodel prediction target. I understand the interest of this choice, but I think it is a strong assumption that should be further discussed in the paper. In particular, the following points are missing:

- Where are the age tracer data? They are not in the Supplementary Material as indicated l. 219, and I could not easily find them in Morgenstern et al. 2018. There is an extensive description of how these data where acquired (l. 228-238) and how they are used to fit LPM (l. 238-264) but results are never presented in the paper while they are very important. Age tracer data fitted by the LPM must appear in Supplementary Material, to evaluate the confidence in the LPM predictions later used.

We thank the reviewer for pointing this out. The majority of, and currently publicly available age tracer data, together with information on tracer inputs, are provided in Morgenstern & van der Raaij (2019). It was an oversight from us to not include the reference to this report, and we will amend the relevant sections accordingly.

Morgenstern U, van der Raaij RW. 2019 Groundwater residence time assessment of Hawke's Bay municipal water supply wells in the context of the Drinking-water Standards for New Zealand. Lower Hutt (NZ): GNS Science. 48 p. https://doi.org/10.21420/8KKH-4W33. 2019.

- Without this, it is hard to evaluate uncertainties associated with the LPM-derived age distributions. Would it be possible to estimate the uncertainties? How much are the trained models sensitive to the LPM? Could uncertainties in LPMs explain part of the errors?

We agree that this would be an interesting topic to study, but this is a separate study in itself. Fitting LPMs to age tracer datasets is subject to several error sources, including the number of different age tracers used, their age and error ranges, the number of measurements at the same XYZ location over time, and the use of binary or single mixing models (dependent on the hydrogeologic system). Therefore, the combination of the errors and how they propagate into the metamodels is quite complex. A paired simple-complex model comparison would be one such methodology that could be used (Doherty & Christenson 2011). We also did touch on in the paper that potentially the metamodelling approaches could be used to identify issues with the LPM interpretations.

Doherty, J. and Christensen, S. 2011. Use of paired simple and complex models to reduce predictive bias and quantify uncertainty. Water Resources Research, 47, https://doi.org/10.1029/2011WR010763.

**1.2.2    Explaining the age-hydrochemistry relationships**

My second main concern comes from the relationships obtained between hydrochemical data and groundwater age distribution and the processes that could explain them.

- Based on which argument and figure can you tell that "NH3-N, Fe and Mn all tend to increase with groundwater age, whereas concentrations of DO and NO3-N tend to decrease" (l. 424)? This affirmation does not appear clearly on Figure 8 and it does not appear clearly either in the correlation matrix Figure 2.

Figure 8 displays the relative weightings of the hydrochemical parameters in the metamodels. It does not, nor is it intended to, display the correlations between the variables.

Figure 2 does indeed show the magnitude and direction of correlations among hydrochemical variables across the whole dataset. For example, DO is seen to be positively correlated with $NO_3$-N ($r = 0.2$) but negatively correlated with Fe ($r = -0.2$), Mn ($r = -0.3$), $NH_3$-N ($r = -0.1$) and $PO_4$-P ($r = -0.4$, mistakenly labelled as DRP, which will be changed). While some of these appear weak, it's because Pearson's r is a measure of linear correlation whereas several of the hydrochemical relationships are known to be non-linear, as shown for the Heretaunga Plains data below (note log scales). The non-linearity in relationships between redox-sensitive parameters is expected given that they tend to be consumed through microbial respiration in a step-wise sequence; for example, $NO_3$-N is usually has to be largely depleted through denitrification before appreciable concentrations of $NH_3$-N build up. We will add a short comment to this effect.

[Figure]

[Figure]

[Figure]

- I found interesting to try to quantify the consumption of DO in the catchment, by assuming that the organic matter oxidation is only related to DO. However, no explanations are given on how the average rate constant was derived and additional information are required. A first-order kinetics on the DOM concentration was considered, therefore not accounting for the DO concentrations (if I understood correctly from the reference given). Is it correct? It should be specified. Which groundwater age percentile was considered for the calculation? How were the DOM concentrations averaged?

We appreciate this interest from the reviewer. Our aim here is just to provide a general indication of the sorts of insights that could be generated on DO consumption rates if better data were available. We caution that our original manuscript acknowledged that the rate of DO consumption can only be evaluated semi-qualitatively at best from the data we have available. We will modify the wording to make this clearer.

We have corrected an error at **Line 445** in the reporting of our estimated rate constant (k). The original manuscript stated that k = -0.6, but actually this should have been log k = -0.6.

To make the calculations easier to follow we will add the first-order kinetic rate expression to the text.

An important point that we didn't make clearly in the original manuscript is that the decline in DO concentration over time depends on the organic matter concentration in the groundwater, either introduced via recharge or acquired during groundwater passage through the aquifer. We will clarify this in the updated manuscript.

An illustrative plot is shown below. It displays the measured concentrations of DO vs the median of the LPM-derived age distribution for each site. Two model curves are shown based on the rate equation now given in the text. The models assume that the governing reaction is $CH_2O + O_2 = H_2O + CO_2$. Both model curves assume the same rate constant (log k = -0.6) and the same initial concentration of DO (8 mg/L), but the top and bottom curves assume different initial organic matter concentrations of 3.3 and 6.6 mg/L, respectively. Note that organic matter concentrations are not typically measured in Heretaunga Plains groundwater or other aquifer systems in New Zealand, so these values were just selected to bracket the dataset – but they are potentially reasonable based on overseas studies.

We welcome advice from the Editor as to whether inclusion of such a plot would be valuable for the manuscript. Our initial sense is that it is based on so many assumptions that it would be better to exclude, but we would welcome feedback.

[Figure]

- The inverse relationship between age and temperature is not expected as we would expect that older groundwater shows higher temperature. But this relationship is really strong and I think this paper would highly benefit from a close look at this relationship and clarifications in the explanations given. I do not understand the calculation of the activation energy made and I doubt the interpretation that is made from it. First, it somehow considers an aggregation of all reaction types. Secondly, where does the k1/k2=0.8 come from? Here, the age ratio is 0.8. But why would the

kinetic rate ratio be equal to the age ratio? I agree that an increase in T would increase the reaction rates. However, how do you relate this to the effect of T on modelled age? Please clarify the process that is presented here to explain the inverse relationship between age and temperature. I would be more convinced by a hydrological explanation. The paper would benefit from a more convincing explanation of the relationship obtained between temperature and age.

We appreciate the reviewer's interest in this finding. Indeed, the strong inverse correlation between T and the modelled groundwater age was a surprising result for us as well.

In the updated paper, we will clarify that the available data from this study do not permit elucidation of the cause(s) of the strong inverse relationship between T and estimated groundwater age, and that the following paragraphs simply present two concepts that could be explored through further investigations.

We feel that the Arrhenius Equation is quite well established and so shouldn't require further explanation in the text, but we will a reference to Langmuir (1997), which contains these same equations.

In the application to the present study, the Arrhenius equation has to be aggregated across all reaction types because there is no available means of identifying specific types of reactions that may be more important than others, or applying the equation to any particular type of reaction. We clarify this in the paper.

The reviewer asks why the reaction rates at two temperatures should be tied to the ratio of ages. This is because we assume that there is a fixed reaction rate constant for each temperature, so a given reaction will proceed at a different rate for the two temperatures being compared. For the reaction to have progressed to the same degree from the same initial chemical condition, our results suggest that the warmer system will take less time. Assuming that the form of the kinetic reaction equation doesn't change (for example it remains first-order), then the ratio of times for the reactions to progress to the same point should be equal to the ratio of their reaction rate constants.

- Relationships between Ca, Mg, Na, K and SiO2 and age would highly depend on the aquifer lithology. Would these elements be better predictors of groundwater age if an *a priori* classification based on the rock lithology was made?

This is a suggestion made by Reviewer RC1, to which we reply in subsection 1.1.2 above. As noted there, we will add an explanation of why we did not apply an a priori segregation of sites based on rock lithology, and we will also add a sentence to our results section that justifies our approach.

**1.2.3    Minor comments**

Fig.1: What are the red lines?

Good find. The red lines identify areas where Morgenstern et al. (2018) found indication that there is no surface water flow contributing to the main aquifers. We will add an explanation to the figure caption.

l. 136: it would be interesting to give the value of the recharge rate of the area

Rakowski & Knowling (2018) estimate the total recharge to the aquifer to be approximately 264 M m3/year, of which losing rivers contribute about 185 Mm3/year, and 79 Mm3/year come from rainfall recharge. We will add this to the updated manuscript version and will also modify the text slightly to make clearer that this is an estimate only.

Rakowski, P. and Knowling, M. J.: Heretaunga aquifer groundwater model: development report, 182 pp., 2018.

Line 140: what is the confined aquifer zone near the coast? Maybe worth showing on the map?

This is a good suggestion. We agree that it could be useful to show the confined aquifer zone on the map. However, the confined aquifer boundary in the Heretaunga Plains is currently in the process of being updated with new data. Alternatively, we could show the currently mapped extents of fine (sand, silt, clay) terrestrial and estuary deposits at the ground surface in the plains if this is seen as useful.

Line 195: there is a confusion between the text and Fig. 3, one refer to mean residence time and the other to the 50$^{th}$ percentile, please correct.

Good catch. We will change this in the text.

Figure 6: At least for the example given in Figure 6, the lumped parameter model should be described in the main text (singular or binary EPMs? Which values of the parameters?)

We agree and will action this in the next version of the manuscript.

Line 337: MAE : Mean Absolute Error?

Good catch. We must have missed explaining this in the paper. In this paper, MAE stands for Median Absolute Error. We will add this to the manuscript.

Figure 8: change DRP for PO4-P as this is how it is referred to in the main text

This will be corrected in the next version of the figure. We have also checked the other figures and have found other instances of this that will also be adapted.

Line 540: I wonder of the generalization of the approach and on the application of the trained model elsewhere. The obtained hydrochemistry-age relationships are not easy to explain (at least for temperature), and therefore it is difficult to tell if they are applicable elsewhere or if they are only related to some local effects. Would other predictive parameters such as depth, distance to the river, or elevation inform on water age predictions?

We agree. We are currently looking into using hydrophysical parameters in addition to the hydrochemistry. This is the plan for the next paper. We will add a sentence about future research in this direction in Section 5.The authors acknowledge that the work might be only applicable to the selected catchment. Is there another similar catchment, where age data are

available and where the models could be applied to determine groundwater age distributions from hydrochemistry, in order to validate the method?

Testing and validating of the models in other catchments is planned for further work.

**1.3 RC3 – Anonymous referee**

Overall, this is an interesting metamodeling application using water quality information to emulate a lumped-parameter model and make forecasts of groundwater age. Two methods were used (gradient boosted regression and symbolic regression) with advantages to each and with generally similar performance. The authors also make a detailed interpretation of the parameter and model behavior.

This is a fine contribution and I have just a few minor comments to consider.

**1.3.1 Minor comments**

Line 61: There is some ambiguity to how the model is described here. It's not really trained on data, but rather is trained on the LPM model that, in turn, is trained on data. Being super clear here is important, particularly for readers less familiar with metamodeling

This is a good suggestion. As noted in our response to RC1, we will clarify this in the updated manuscript.

Figure 1 and in the text: The clusters from previous work are both identified on the figure and in the text, but no context is provided beyond a reference to previous work. A sentence or two would be key to explain this.

We use the clustering from previous work for two purposes. The first purpose is just to simplify the description of hydrochemical variations across the Heretaunga Plains aquifer system. We will add a sentence to explain this rationale at **Line 168**.

The second purpose we use the clustering for is to test whether the age models are able to perform adequately on all groundwater chemical categories, but without have pre-segregated the dataset and training separate machine learning models for each cluster. Please note our responses to similar comments from Reviewer RC1 in subsection 1.1.2 above. As noted there, we will add an explanation of why we did not apply an a priori segregation of sites based on rock lithology at **Line 266**, and we will add a sentence to our results section at **Line 378** that justifies our approach.

Figure 2 and elsewhere: Many of these water quality constituents are obviously identified by their chemical formulae, but some of not defined. Even if it's in supplemental material, a table defining the quantities would be helpful.

We agree that this is an oversight from us. We will add a table to the supplemental material as suggested and provide a reference to the table in the text.

| Abbreviation | Name | Units |
| --- | --- | --- |
| Ca | Calcium | mg/L |

| | | |
|---|---|---|
| Mg | Magnesium | mg/L |
| Na | Sodium | mg/L |
| K | Potassium | mg/L |
| $HCO_3$ | Bicarbonate | mg/L |
| Cl | Chloride | mg/L |
| $SO_4$ | Sulphate | mg/L |
| $NO_3$-N | Nitrate-nitrogen | mg N/L |
| $NH_3$-N | Ammoniacal nitrogen | mg N/L |
| $PO_4$-P | Phosphate-phosphorus | mg P/L |
| Fe | Iron | mg/L |
| Mn | Manganese | mg/L |
| $SiO_2$ | Silica | mg/L |
| B | Boron | mg/L |
| F | Fluoride | mg/L |
| pH | pH | pH units |
| EC | Electrical conductivity | $\mu$S/cm at 25°C |
| DO | Dissolved oxygen | mg/L |
| T | Temperature | °C |
| $\delta^{18}O$ | Oxygen-18 | per mil (‰) |
| $\delta^2H$ | Deuterium | per mil (‰) |

Line 290: There seems to be a formatting glitch here – hard to understand what the equation is meaning to explain.

Good catch. Symbols must have been converted between versions. As already noted in our response to RC1, we will correct this.

Line 327: more formatting glitches

Another good catch. Will correct this.

Lines 359-362: This is a great point and I appreciate the context because it's true that the extrema of the distribution would be of interest to many users.

We thank the reviewer for support of this point made in the original manuscript.

**1.4     RC4 – Anonymous referee**

This manuscript aims at assessing the validity of using two machine leaning techniques to extrapolate beyond available groundwater age data and infer the lumped RTD from hydrochemistry.

This contribution is novel and appears quite appealing to complement tracers dataset which are costly and time consuming.

The manuscript is nicely written and easy to follow.

**1.4.1   Training chemistry-based metamodels on LPM-derived age distributions**

I have reservations about the choice of the LPM models as calibration targets. I understand that the study is closer to the reality in which the age distribution is unknown. Still, I consider that it would have been much stronger to test the validity of the methodology on a pure synthetic case controlling every aspect of the problem: data and associated uncertainty, full shape of the age distribution, etc. An important aspect as well is that, without a priori information about the age distribution, a few LPM differing in their hydrogeological conceptual representation can equally fit. My point is that it is difficult to evaluate the validity of a calibration or inference methodology on real largely under-constrained cases. One way to tackle this would be I think to highlight the fact that the system studied here is a "not so complex" system (a textbook system?) and have been widely studied so that the target LPM is a more than reasonable estimation (see my minor comment below).

Please refer to our response to Reviewer RC1, given in subsection 1.1.1 above. As listed there, we have made several modifications to the manuscript to clarify and justify our approaches.

The reviewer suggests that we could have conducted a purely synthetic study based on models in which all parameters and processes were fully constrained. We did consider this idea and may pursue it in the future, but we ruled it out for the current investigation because we considered the chemical process datasets and models to be too uncertain to be useful. More detail is on this below.

In order to undertake a purely synthetic study we would need to develop a model of the groundwater flow and transport system. Much of this modelling has already been undertaken. Simulations are already possible for tritium transport and particle tracking or direct age simulation to enable the age distributions of groundwater to be evaluated spatiotemporally across the model domain. Further work is underway to improve the existing groundwater flow and transport models but is not yet published.

In order to undertake a purely synthetic case study we would also need to implement a model of the chemical evolution of groundwater over time. One option would be to use a forward simulation based on lab-derived reaction rates, e.g. using a programme such as PHREEQC, Geochemist's Workbench, etc. The other option would be to apply a reaction rate model based on field observations, such as PROFILE or ForSAFE (see Sverdrup et al. 2019, cited in the original manuscript). In either case, we would need to be certain that these models contain geochemical processes that are relevant to our particular field location.

But just as important, regardless of which type of geochemical reaction model was selected, we would have the major challenge that we do not have data including but not limited to: 1) the mineralogical composition of the aquifer materials or how they vary spatially; 2) the key factors such as reactive surface area, which control water-rock reaction rates; 3) microbial processes and their rates and spatial variability; or 4) dissolved and solid phase organic matter concentrations and their reactivities.

The result is that we felt that we could conceivably develop a groundwater flow and transport model with accompanying estimates of age distributions, but our ability to model the geochemistry would be too underconstrained to be useful.

**1.4.2    Chaining approach**

I have reservations as well about the independence for the percentiles and the further chaining approach. It appears to me that it goes again physics and flow mechanics to consider percentiles as separate entities, and not the age distribution as a whole. My point is that a LPM or numerically-generated distribution lies on a hydrogeological conceptual representation which describes the functioning of the system. It has been shown (Leray et al, 2019: https://doi.org/10.1016/j.jhydrol.2019.04.032) that local modification of the system properties affects not only local flow lines and mass balance locally but the overall response and functioning of the system and consequently the age distribution. So it is confusing to me that the distribution is considered by part (even if the chaining approach intends to reconstruct the puzzle)

We thank the reviewer for this insightful comment. We agree that the percentiles in a single age distribution must have a mathematical relationship driven by the groundwater flow regime. The approach that we took in the manuscript should not be taken as a disagreement with this statement – rather, our approach was followed to test the appropriateness of using LPM-derived age distributions as our modelling objective.

We initially constructed unchained models for the *individual* percentiles as a means of testing the validity of the shapes of the age distributions produced by the LPMs. By modelling one percentile at a time, we aimed to determine whether there were any sites for which the SR or GBR models produced misfit, which may have demonstrated that the shape of the LPM age distribution was inappropriate at those sites. Then by comparing the age estimates derived for different percentiles (i.e. from different unchained SR or GBR models) at single site, we could diagnose whether the LPM mean age was erroneous (as shown by systematically incorrect age estimates for all percentiles), or the LPM's age distribution had the wrong shape (as shown by different misfit for different percentiles), or a combination of both.  We will clarify this at **Line 281**.

The reason for subsequently constructing the chained models was already described in the original manuscript (at **Line 302**): "[Chaining] was done to ensure that the separately simulated percentiles had an appropriate relationship to each other, e.g., that the value for the 10th percentile in the age distribution for any sample had to be greater than or equal to the 5th percentile in the age distribution at the same sample."

Our results section already discussed the insights that could be gained by comparing the quality of SR and GBR model fits across sites and percentiles. We will add a sentence at **Line 378** to explain that "there were few sites for which clear errors in the shape of the LPM-derived age distribution could be identified based on differences in the quality of fit of unchained model fits across different percentiles, so we conclude that the LPMs applied in this investigation are generally appropriate to represent the age distributions in the study area."

**1.4.3    Minor comments**

Line 26: I would write the age as plural ("understanding the ages of water") to reinforce the fact that natural groundwater systems are made of a wide variety of flow paths and consequently of residence times (or ages). If it is correct grammatically of course.

Agreed. Will modify text as recommended.

Line 53: "most such previous studies". Revise

No change will be made in response to this comment. Perhaps we are being slightly pedantic, but we have opted to keep the original wording. The use of the term 'such previous studies' is to show that we are referring to the 'various less time and cost-intensive methods have previously been trialled to increase the amount of available groundwater age data in areas where no age tracers have been sampled' – in other words, we are not referring to all studies about groundwater age.

Line 218: I am not an expert but should it not be half of the detection limit?

We will clarify the text at **Line 217** to say "Censored and uncensored results below the highest censoring threshold for each parameter were replaced with the corresponding analytical detection limit (Helsel et al., 2020)". The reason for this approach is that it isn't possible to tell the difference between concentrations reported (for example) as <0.05, 0.03, <0.06 so they should all be considered equivalent and set at 0.06 for SR and GBR model training.

Lines 252 to 254: It is argued that the EPM provided good matches for a wide range of New Zealand systems. A fault-bounded, local, relatively homogeneous and thick system with uniform recharge rate upstream and zero recharge rate downstream looks like an EPM to me. So, I think the validity of the EPM should be argued considering specific aspects of the system (that may be quite similar to other sites in New Zealand)

We agree that this is an important aspect to include. We will add a sentence to clarify that the EPM does indeed match the hydrogeological system, at **Line 254**.

Line 278: to differentiate.

Good catch. Will be corrected.

---

## Referee Report (RR1)

**General comment**

The authors have significantly reworked their manuscript, clarifying many points, and streamlining some of the analysis. I still think that the manuscript is too long, especially the introduction (that reads like a review), and that the case study chosen is too complex and fraught with specificities, but this may be a question of taste.

There is however one aspect of the analysis I disagree with. The authors present the metamodelling as successfully reproducing the reference transit time distributions obtained from tracer data using lumped-parameter models. But I do not think that figure 6 conveys this at all. On the top two figures, the range of transit times estimated for the $90^{th}$ and the $95^{th}$ percentiles is enormous, and hence obviously not constrained well by the chemical data. The bottom left figure on the contrary clearly shows that the metamodel manages to approach the reference distribution. But the bottom right figure illustrates another problem, namely that the metamodels can systematically bias the entire transit time distribution to lower values. So maybe the metamodelling works well on average, as seems to be indicated by the statistics presented, but there are still serious problems in some cases, and I feel this is being sligthly swept under the carpet. Further below in the discussion (L688), the authors suggest that this deviation might in fact indicate that the calibrated transit time distribution is inappropriate, and that using groundwater chemistry data together with a metamodel could guide model choice for the lpm. This is interesting, but still a bit too hypothetical.

I understand that the authors see this manuscript as a proof of concept, but I cannot help thinking that exploring first what is only been hinted at in the discussion, and then presenting a series of simple case studies illustrating clearly the advantages and the problems of the proposed metamodelling instead of suggesting what one could do with it would have made the method the authors propose more concise, much clearer and to the point.

---

## Author Response (AR2)

We thank the reviewer for their insightful comments which have touched upon topics that have been discussion points between authors. The reviewer's comments have helped us refine and clarify the narrative of this manuscript. Our responses to the reviewer's comments (in black) are in blue below.

General comment

The manuscript presents the use of two machine learning algorithms to estimate groundwater transit time distributions as complement to or in the absence of appropriate age dating tracers.

The central research question, the state of the art and the local setting are all very competently presented.

The analysis itself however seems to me still incomplete and rather unfocused. While the authors discuss at length the effect of different processes and parameter choice on estimate results, they only pay lip service to what should be the central step, namely quantifying the deviation and bias of the estimates obtained from the metamodels compared to the reference transit time distributions. I think that simply showing the transit time distributions and then declaring that the fit is overall satisfactory is not convincing enough. Furthermore, the discussion on the results of the case study are too detailed and specific, and dilute the manuscript instead of keeping it compact and to the point.

We thank the reviewer for this comment, which provided an opportunity to better clarify the purpose of the analysis. The purpose of the analysis was to explore in a 'proof of concept' sense the effectiveness of data-driven or machine learning algorithms to the estimation of groundwater age on the basis of water chemistry data. Because the groundwater age data was itself an estimate based on a very simple representation of the physical system (e.g. the Lumped Parameter Model 'LPM') we decided that defining the deviation and bias from this estimate of groundwater age was of questionable value in these experiments. We agree that this output would be useful if the physical system was simulated in greater detail, and this is the focus of our current work.

The authors' suggestions as to the further use that could be made of the metamodel approach presented are interesting and seem potentially useful indeed, but I would have liked to see some concrete examples.

We agree that concrete examples of the use of metamodel approaches in water management will provide better context for the utility of this work. As the reviewer notes we had some examples in the results and discussion section. We have added some additional text to add further conceptual examples to the manuscript to provide further context.

This includes discussion of the insights that groundwater age can provide into how groundwater systems function, and how they may be changing over time. These estimated ages can also provide history matching targets for numerical models used to inform groundwater allocation decisions.

We have not included any numerical examples due to the length of the paper, and the intention to fully describe concrete examples in future publications.

All in all, the approach seems interesting, but the analysis itself would gain in depth by (i) refocusing it on the central question of how well the metamodels perform, (ii) reorganising and cutting down the discussion and (iii) potentially illustrating what is only hinted at in the concluding section.

We thank the reviewer again for raising these issues. In terms of the three points raised above we have done the following:

(i) Performance: We have replaced concepts of 'fit' with 'correspondence' to avoid ambiguity around any claims to fit a 'truth'. This makes the proof of concept focus of the paper more clear. This also avoids erroneous extrapolations given the fact we are fitting to a simple model which is also accompanied by simplification errors that have not been explored in this work.

(ii) We have reorganised and streamlined the discussion. We have retained sections that we believe are critical to highlight the additional insights and potential value of this type of meta modelling approach. We are hesitant to remove more of the discussion at this stage of the review process, as four other reviewers have not raised this concern.

(iii) We have added a brief discussion of how meta modelling outputs can be used to inform environmental management decisions.

Specific comments

L36: The parameters of a lumped parameter model are sometimes estimated from hydrogeological information rather than from tracer data (see for instance Abrams and Haitjema, Ground Water 56 (3), or Bailleux et al., Hydrogeology Journal 23 (7)). I think this point should also be taken up in the discussion (L616).

Amended text. We agree with the reviewer and have added a comment on this into the introduction. However, we did not revisit this in the discussion as that may distract from the 'proof of concept' focus of the paper (e.g. to estimate age on the basis of water chemistry).

L209: Do you mean median of MEAN residence times, or the median of the distribution of individual flow lines' residence times ?

Amended text. We meant median of the mean residence times and have adjusted the manuscript to clarify this.

L284: I do not understand why you write that the cause for the lack of fit of a single EPM to tracer data indicates that the transit time distribution has "changed over time". Isn't it rather because the real transit time distribution deviates significantly from the exponential model, as could be expected for more complex hydrogeological settings ?

We amended the text to clarify our meaning and address any confusion.

In this case, we are not talking about e.g., EPM vs BMM, but rather this is referring to different samples taken over time at the same site. The multi-age tracer long-term data appear to fit two different age distribution modes, one with a younger, and one with a slightly older age distribution. Such bi-modal age distributions are plausible, with the potential reason for this being seasonal changes, for example increased pump rates during summer, and increased recharge rates during winter. Therefore, we fitted two different age distributions to the data,

one matching the data indicative of younger water, and one matching the data indicative of older water.

L325: By "appropriate", do you mean that the choice of these criteria has been tested in any way, or is it simply that this is what everyone in that field does, hoping for the best ?

Amended text. We agree the word 'appropriate' is ambiguous and so we have removed it. It is indeed standard practice, but there are issues with it as discussed in Schöniger et al. (2014) as cited.

L365: The matches shown in figure 6 and in the supplementary figures S1 to S4 are sometimes close and sometimes not. The widths of some estimated ensemble percentiles are also huge in some cases. So overall, unless you find a way to quantify the deviations between LPM transit time distributions used as reference, and estimated percentiles, I would be much more guarded in the assessment of goodness of fit.

Amended text. We agree with the reviewer. We are focussing on the broad ability of the metamodels to represent a LPM derived distribution. To better convey the 'proof of concept' focus of the paper we have amended the discussion, adopting the term 'correspondence' rather than 'fit' between LPM transit time distributions and those derived from the metamodels.

We also acknowledge the heterogeneity in the system that cannot be presented by an LPM, which may compromise the age estimation, and have also mentioned this in the text.

L384-391: The paragraph starting with "This finding […]" may be moved to the discussion, I think.

We are unsure of the reviewers meaning here as this section is already in the Results and Discussion section.

L415: Since the core of the manuscript is to test whether machine learning can be used to estimate transit time distributions from hydrochemical datasets, I think you need to include a way to quantify goodness of fit and deviations from the reference transit time distributions. Simply relying on a graphical comparison, and then declaring that the fits are good enough seems very unsatisfactory to me.

Amended text. We agree with the reviewer. Goodness of fit information was provided in the Supplementary material Table S2, and we have amended the manuscript text to make this clearer.

We also acknowledge that the goodness of fit metrics are less appropriate in this study, where the LPM estimates are also accompanied by a model simplification error. We note that a paired simple-complex model intercomparison could be undertaken to better estimate model structural errors associated with the LPM in this context (Doherty and Christensen 2011 discuss this method).

We have amended the text further, so that we discuss correspondence between the metamodel and LPM outputs, rather than 'fit'. However we retain the standard goodness of fit metrics in the SI, as a metric of similarity rather than ability, in a heuristic sense.

L424: The phrasing "can successfully be used to predict groundwater age distribution" is too much of a statement to be taken at face value instead of the result of a thorough analysis of goodness of fit (see preceding comment).

Amended Text. We agree and have amended the sentence. This also links to the response above to L415.

L427: The sensitivity analysis is nice, but should not replace the much more central step of quantifying the deviation and the bias between SR and GBR models and the reference distribution. In my opinion, this step is still largely missing from the overall analysis.

We thank the reviewer for this comment. In future work we are looking at characterising bias and deviation of LPM models using numerical experiments where a synthetic truth can be used as an objective reference (this would address the fact that we don't have 'real' age data, only interpreted age). At this point we believe that analysis of metamodel bias and deviation would be useful. However, we believe that this analysis goes beyond the scope of this paper which is to explore whether or not chemistry-age relationships are sufficiently strong to support metamodels.

L570: I am surprised that seasonal variations in groundwater heads would be so large as to affect the calculation of mean residence times in such a wet environment as New Zealand, at least for aquifers that are not largely fed from infiltrating streams.

L583: Same remark as above concerning water quality. How come water quality is so variable and dependent on sampling season for groundwater environments ?

This response is in regard to the reviewer's comments on L570 and L583.

Most aquifers in New Zealand receive a significant proportion of recharge from infiltrating streams. For the particular case study, this is estimated to be more than 66% of the mass balance (as discussed in section 2.2 of the paper).

Rainfall, evapotranspiration, as well as water use, vary a lot between seasons in New Zealand. As a result, the rainfall and river recharge patterns and amounts may vary, which can affect groundwater chemistry and groundwater age distributions in the aquifer.

While this is generally the exemption, usually, shallow unconfined wells are affected, which, in winter, tap into the fresh local recharge pulse (young water). In contrast in summer, the fresh winter recharge pulse becomes depleted and, as a result, the discharge from the well at lower water levels reflects only the deeper older groundwater. This seasonal variability is clearly indicated by different age tracer concentrations at different seasons, and if the older and shallower water contrast in hydrochemistry, also hydrochemistry would vary.

Not much is known yet internationally about such seasonal variabilities of groundwater age in wells. It is especially in the southern hemisphere (and particular in NZ with its high-resolution tritium input available), where due to the absence of 'bomb-tritium' young and older water show a large contrast in tritium concentrations, that such effects can easily be studied. However, this is not the subject of this paper.

L598: I would use the word "reference" rather than "truth", even in brackets.

Good point. Amended.

L607-609: I completely agree.

---

## Author Response (AR3)

Reviewer's comment:

**General comment**

The authors have significantly reworked their manuscript, clarifying many points, and streamlining some of the analysis. I still think that the manuscript is too long, especially the introduction (that reads like a review), and that the case study chosen is too complex and fraught with specificities, but this may be a question of taste.

There is however one aspect of the analysis I disagree with. The authors present the metamodelling as successfully reproducing the reference transit time distributions obtained from tracer data using lumped-parameter models. But I do not think that figure 6 conveys this at all. On the top two figures, the range of transit times estimated for the 90th and the 95th percentiles is enormous, and hence obviously not constrained well by the chemical data. The bottom left figure on the contrary clearly shows that the metamodel manages to approach the reference distribution. But the bottom right figure illustrates another problem, namely that the metamodels can systematically bias the entire transit time distribution to lower values. So maybe the metamodelling works well on average, as seems to be indicated by the statistics presented, but there are still serious problems in some cases, and I feel this is being sligthly swept under the carpet. Further below in the discussion (L688), the authors suggest that this deviation might in fact indicate that the calibrated transit time distribution is inappropriate, and that using groundwater chemistry data together with a metamodel could guide model choice for the lpm. This is interesting, but still a bit too hypothetical.

I understand that the authors see this manuscript as a proof of concept, but I cannot help thinking that exploring first what is only been hinted at in the discussion, and then presenting a series of simple case studies illustrating clearly the advantages and the problems of the proposed metamodelling instead of suggesting what one could do with it would have made the method the authors propose more concise, much clearer and to the point.

Authors' response:

We are pleased to have the opportunity to further improve our manuscript.

As the reviewer points out, it is complex! Unfortunately, this complexity is a reality for applications in dynamic real-world systems.

The remaining concerns of both the reviewer and the editor appear to relate to the presentation of the age distributions in Figure 6 (and S1-S4), and the associated text. We want to clarify that our choice of sites to show, as examples, in Figure 6 was designed to highlight the variation in model results; these sites definitely do not represent the site with the "best" correspondence. We refute the notion that the poorer results have been "swept under the carpet". However, we acknowledge that the text around these results may not have highlighted the variation in performance as we intended. We have reworded and slightly restructured Section 4.1 to try and make this clearer.

We hope that this reworking provided more clarity to the presentation of the predicted age distributions, including the coherency (or otherwise) of the prediction median absolute deviations.